# Regularized Offline Policy Optimization with Posterior Hybrid Bayesian Belief

**Hongqiang Lin** [1 2]  **Pengfei Wang** [3]  **Nenggan Zheng** [1 2 4]

## Abstract

Offline reinforcement learning (RL) aims to optimize policies from pre-collected datasets. A bottleneck of this paradigm is managing epistemic uncertainty, which arises from limited data coverage (sample-level) and the ambiguity in identifying transition dynamics from finite data (model-level). To provide a unified quantification of these uncertainties, Bayesian RL has been proposed by treating the dynamics model as a random variable and maintaining a corresponding belief. Despite its theoretical appeal, policy optimization in Bayesian RL remains computationally challenging as it requires solving composite objectives with expectations. Prior methods either employ search-based techniques with poor computational scalability or impose restrictive posterior assumptions that sacrifice the adaptability of Bayesian RL. To address these limitations, we propose Posterior Hybrid Bayesian Belief (PhyB), which reformulates the expectation as a convex combination over a subset of dynamics models. Theoretical analysis demonstrates that the objective discrepancy induced by this approximation remains bounded. Based on PhyB, we develop an iterative regularized policy optimization algorithm that provides metric-agnostic guarantees for monotonic improvement until convergence. Empirical results demonstrate that PhyB achieves state-of-the-art performance on various benchmarks.

## 1. Introduction

Offline RL aims to optimize policies from pre-collected datasets (Levine et al., 2020), avoiding the safety risks and costs associated with environmental interaction in online RL (Sutton & Barto, 2018; Gottesman et al., 2019; Kumar et al., 2022). The primary challenge in offline RL is distribution shift (Kumar et al., 2020; Yang & Wang, 2025). Within the framework of uncertainty quantification, extrapolation errors caused by distribution shift in out-of-distribution (OOD) regions is highly correlated with epistemic uncertainty (Lu et al., 2022). Consequently, quantifying this uncertainty is critical for achieving robust performance in the absence of online environment interaction.

Epistemic uncertainty in offline RL can be categorized into two levels. The first is sample-level uncertainty, which arises from partial dataset coverage (Rigter et al., 2023). The second is model-level uncertainty, which arises from the inability to uniquely identify the underlying transition dynamics from finite datasets (Ghosh et al., 2022). Existing model-based algorithms utilize the standard deviation of ensemble predictions to quantify the sample-level epistemic uncertainty and penalize OOD value estimates (Yu et al., 2020; Sun et al., 2023; Zhai et al., 2024; Qiao et al., 2025; Lin et al., 2025). Despite their empirical success, these methods optimize the dynamics model and the policy under independent objectives, which prevents the integration of model-level uncertainty into policy optimization.

Bayesian RL (Ghavamzadeh et al., 2015) provides a framework to unify various forms of uncertainty within the offline RL. By treating dynamics models as random variables and maintaining a corresponding posterior belief distribution over the model space, Bayesian RL allow for simultaneous quantification of model-level uncertainty through Bayesian inference and sample-level uncertainty through prediction.

Policy optimization in Bayesian RL is typically formulated as a composite optimization problem involving expectations over model posteriors, which are often intractable in continuous spaces (Jiang et al., 2025). Although prior methods utilize solvers such as mixed-integer linear programming (Lobo et al., 2020) or Monte Carlo tree search (Rigter et al., 2021), these methods fail to scale to high-dimensional tasks. Consequently, current methods resort to fixed posterior assumptions or simplified formulations (e.g., robust MDP) to maintain tractability (Rigter et al., 2022; Dong et al., 2024). However, these oversimplifications inevitably compromise the inherent performance and adaptability of Bayesian RL.

---

[1]College of Computer Science and Technology, Zhejiang University, Hangzhou, China [2]Qiushi Academy for Advanced Studies, Zhejiang University, Hangzhou, China [3]School of Software Technology, Zhejiang University, Ningbo, China [4]School of Computer and Information Engineering, Bengbu University, Bengbu, China. Correspondence to: Pengfei Wang <pfei@zju.edu.cn>.

*Proceedings of the 43$^{rd}$ International Conference on Machine Learning*, Seoul, South Korea. PMLR 306, 2026. Copyright 2026 by the author(s).

Therefore, a fundamental question remains to be addressed: *Can we develop a computationally tractable offline policy optimization algorithm under Bayesian beliefs without relying on restrictive assumptions on the posterior distribution?*

We give an affirmative answer to this question by proposing Posterior Hybrid Bayesian Belief (PhyB) and an iterative regularized policy optimization algorithm. Our main contributions are summarized as follows:

**Belief Formulation.** PhyB formulates transition dynamics as random variables rather than point estimates. Unlike prior works, PhyB avoids restrictive assumptions on the posterior distribution and maintains computational efficiency by reformulating the expectation as a convex combination over a subset of dynamics model.

**Iterative Regularized Policy Optimization.** To account for the posterior geometry while maintaining computational tractability, we decompose the optimization task into a sequence of Bregman-regularized subproblems and propose an iterative algorithm to solve these subproblems.

**Theoretical Analysis.** Our theoretical analysis illustrates two primary properties of PhyB. First, the objective discrepancy arising from the convex combination reformulation is bounded. Second, PhyB induces pessimism that is both controllable and monotonic with respect to key hyperparameters. Policy optimization under PhyB provides metric-agnostic guarantees for monotonic improvement until convergence.

Empirically, our method achieves superior performance on D4RL (Fu et al., 2020) and stochastic benchmarks with fixed hyperparameters. The code is available at https://github.com/HQ-Lin/PhyB.

## 2. Preliminaries and Notations

### 2.1. Markov Decision Process (MDP) and Bayes-Adaptive MDP (BAMDP)

A standard MDP is represented by the tuple $\mathcal{M} = (\mathcal{S}, \mathcal{A}, \tau, r, \rho_0, \gamma)$, comprising the state space $\mathcal{S}$, the action space $\mathcal{A}$, the transition dynamics model $\tau : \mathcal{S} \times \mathcal{A} \to \Delta(\mathcal{S})$, where $\Delta(\cdot)$ denotes the probability simplex, the reward function $r : \mathcal{S} \times \mathcal{A} \to [-R_{\max}, R_{\max}]$, initial state distribution $\rho_0$, and discount factor $\gamma$. The objective is to learn a policy $\pi : \mathcal{S} \to \Delta(\mathcal{A})$ that maximizes the expected return:

$$\eta(\pi, \tau) = \mathop{\mathbb{E}}_{\rho_0, \pi, \tau} [\sum_{t=0}^{\infty} \gamma^t r(s_t, a_t)]. \tag{1}$$

In the Bayesian RL framework, epistemic uncertainty about the environment is modeled by treating transition dynamics as random variables. This formulation induces a Bayes-Adaptive MDP (BAMDP), characterized by the tuple:

$$\widetilde{\mathcal{M}} = (\mathcal{S}, \mathcal{A}, \mathbb{P}, r, \rho_0, \gamma),$$

where $\mathbb{P}$ represents the probabilistic belief over the space of candidate dynamics model. Historically, capturing this belief involved maintaining distributions over the weights of a function approximator (e.g., Bayesian neural networks). However, the high dimensionality of deep neural networks renders exact inference computationally intractable. To reconcile Bayesian theory with deep RL scalability, existing methods approximate the posterior directly over the transition dynamics $\tau$ rather than the network weights (Chua et al., 2018; Ghosh et al., 2022; Rigter et al., 2022; 2023).

### 2.2. The Application of Bayesian RL in Offline Setting

Bayesian RL captures epistemic uncertainty by modeling dynamics as a random variable governed by a belief distribution. A common application of this framework in offline settings is quantile optimization:

$$\max_{\pi} [\min_{\tau \in \mathcal{T}}]^k \eta(\pi, \tau), \tag{2}$$

where $\mathcal{T} = \{\tau_0, \tau_1, \cdots, \tau_{N-1}\}$ denotes the model ensemble with size $N$, operator $[\min_{\tau \in \mathcal{T}}]^k f(\tau)$ represents the $k$-th minimum of $f(\tau)$ over $\tau \in \mathcal{T}$. The optimization in Eq. (2) can be reformulated as maximizing the $\frac{k}{N}$-quantile of the objectives $\{\eta(\pi, \tau)\}_{\tau \in \mathcal{T}}$. Notably, when $k = 1$, Eq. (2) reduces to the optimization objective of robust MDP, which is known to exhibit over-conservatism and may result in suboptimal performance (Dong et al., 2024).

From the perspective of Bayesian RL, standard quantile optimization implicitly assumes a Dirac likelihood over the model ensemble $\mathcal{T}$, concentrating all probability mass on a single model $\tau_*$ at the $\frac{k}{N}$-quantile while disregarding all others. This point-mass assumption can be relaxed by replacing the Dirac likelihood with a unimodal distribution centered around a specific quantile (Guo et al., 2022).

Point-mass and unimodal likelihoods fail to fully utilize the information contained within the model ensemble. The available levels of pessimism are limited to a discrete set of quantile anchors (e.g., $\{\frac{1}{N}, \ldots, \frac{N-1}{N}, 1\}$). This restriction prevents the continuous modulation of pessimism and fails to capture optimal dynamics that lie between these discrete points, ultimately hindering the accuracy of uncertainty quantification and the capacity for generalization (Choi et al., 2024). To address these limitations, we propose a method that aggregates multiple unimodal distributions into a single hybrid multimodal posterior, which allows the continuous and data-driven adaptation of the pessimism level within a closed interval.

# 3. Posterior Hybrid Bayesian Belief (PhyB)

Applying Bayesian RL methods to the offline setting requires satisfying two conditions: (1) accurately quantifying epistemic uncertainty; (2) ensuring pessimism, such that the policy's expected returns under the MDP sampled from the posterior belief lower bound the true returns. This section presents the formulation of Posterior Hybrid Bayesian Belief (PhyB) to address these challenges and provides a comprehensive analysis of its theoretical properties.

## 3.1. Formulation

We begin with a prior belief $\mathbb{P}(\tau)$ representing our initial knowledge of the environment. Our goal is to derive a reliable posterior distribution $\widetilde{\mathbb{P}}(\tau)$ from which dynamics models are sampled for policy optimization. The objective function is defined as follows.

**Definition 1** (Objective Function). We define cumulative discounted return as $\mathcal{R}_{s,a} = \sum_{t=0}^{\infty} \gamma^t r(s_t, a_t)$. The objective function is defined as:

$$\eta(\pi) = \underset{\substack{\rho_0, \pi \\ \tilde{\tau}_0 \sim \widetilde{\mathbb{P}}(\tau)}}{\mathbb{E}} \left[ \underset{\substack{\tilde{\tau}_0, \pi \\ \tilde{\tau}_1 \sim \widetilde{\mathbb{P}}(\tau)}}{\mathbb{E}} \left[ \cdots \underset{\tilde{\tau}_\infty, \pi}{\mathbb{E}} \left[ \mathcal{R}_{s,a} \right] \right] \right]. \quad (3)$$

To evaluate Eq. (3), we define theoretical Bellman evaluation operator $\hat{\mathcal{B}}^\pi$:

$$\hat{\mathcal{B}}^\pi Q(s, a) = r(s, a) + \gamma \underset{\substack{\tau \sim \widetilde{\mathbb{P}}(\tau) \\ s' \sim \tau, a' \sim \pi}}{\mathbb{E}} [Q(s', a')]. \quad (4)$$

However, maintaining a posterior over the continuous model space through inference is computationally intractable. We address this by constructing a pessimistic subset and maintaining a belief over it to approximate the inference process.

**Definition 2** (Pessimistic Subset). Given a finite model ensemble $\mathcal{T} = \{\tau_0, \ldots, \tau_{N-1}\}$ of size $N$, where each $\tau_i$ is sampled i.i.d. from the prior $\mathbb{P}(\tau)$, the pessimistic subset $\widetilde{\mathcal{T}} \subseteq \mathcal{T}$ is formed by selecting the $k$ models that correspond to the bottom-$k$ $q_{\tau_i}(s, a) = \mathbb{E}_{\tau_i, \pi}[Q(s', a')]$ values.

We assign a weight to each model in the pessimistic subset $\widetilde{\mathcal{T}}$ and consider their convex combination. To enforce pessimism, models with lower Q-values should be assigned higher weights. To preserve ensemble diversity and avoid collapsing onto a single model, we present an entropy-regularized formulation as follows:

$$\min_\alpha \sum_{i=0}^{k-1} \alpha_i q_{\tau_i}(s, a) + \lambda \alpha_i \log \alpha_i. \quad (5)$$

The optimal solution for Eq. (5) is given by $\alpha_i \propto \exp(-\frac{1}{\lambda} q_{\tau_i}(s, a))$, with the full derivation provided in the Appendix B.

We construct a posterior belief $\widetilde{\mathbb{P}}(\tau)$ over the model ensemble such that $\mathbb{E}_{\widetilde{\mathbb{P}}(\tau)}[q_\tau(s,a)] = \sum_{i=0}^{k-1} \mathbb{E}_{\mathbb{P}(\tau)}[\alpha_i q_{\tau_i}(s,a)]$. This is achieved by reweighting the prior with a likelihood ratio $\xi(q_\tau)$. Under this principle, computing the expectation under the posterior belief is equivalent to evaluating a weighted expectation under the prior distribution.

**Proposition 1** (Likelihood Ratio). *Let $N$ and $k$ denote the sizes of the model ensemble $\mathcal{T}$ and pessimistic subset $\widetilde{\mathcal{T}}$, respectively. Let $\mathcal{F}$ denote the cumulative distribution function of $q_\tau(s, a)$. Then the likelihood ratio is given by:*

$$\xi_N(q_\tau) = \sum_{i=0}^{k-1} \alpha_i \frac{N!}{i!(N-i-1)!} \mathcal{U}(\mathcal{F}(q_\tau)),$$

*where $\mathcal{U}(\mathcal{F}(q_\tau)) = [\mathcal{F}(q_\tau)]^i [1 - \mathcal{F}(q_\tau)]^{N-i-1}$, and the weights $\{\alpha_i | \sum_{i=0}^{k-1} \alpha_i = 1, \alpha_i \geq 0\}$ are the solution to the optimization problem in Eq. (5).*

We observe that for each $i \in \{0, \cdots, k-1\}$, the term $\mathcal{U}(\mathcal{F}(q_\tau))$ is a unimodal function. This function attains its maximum value at the $\frac{i}{N-1}$-quantile, and the corresponding maximizer $q_{\tau^*}$ satisfies $\mathcal{F}(q_{\tau^*}) = \frac{i}{N-1}$. Our posterior $\widetilde{\mathbb{P}}(\tau)$ is thus a hybrid distribution constructed from this family of unimodal functions. We refer to this formulation as Posterior Hybrid Bayesian Belief (PhyB).

**Remark** (Asymptotic Behavior). Proposition 1 characterizes the likelihood ratio $\xi_N(q_\tau)$ for a finite ensemble size $N$. Theoretically, the weighted sum in our objective constitutes an *L-statistic* (linear combination of order statistics). As $N \to \infty$, noting that the discrete weights $\alpha_i$ converge to a continuous spectral function $J(u) \propto \exp\left(-\frac{\mathcal{F}^{-1}(u)}{\lambda}\right)$ (i.e., $\alpha_i \approx \frac{1}{N} J(i/N)$), the finite-sample estimator converges almost surely to the expectation under a limiting posterior. In this limit, the mixture of Beta distributions in $\xi_N(q_\tau)$ simplifies to the functional form $\xi_\infty(q_\tau) = J(\mathcal{F}(q_\tau))$, which exponentially reweights the prior towards lower Q-values. This implies that PhyB serves as a statistically consistent approximation to a target distribution shift defined by the score function $J$.

**Remark.** As established by Proposition 1, the likelihood depends directly on $\{q_{\tau_i}(s, a)\}_{i=0}^{k-1}$, which are implicitly contingent upon the offline dataset $\mathcal{D}$. This dependency ensures that the posterior update is intrinsically tied to the empirical data distribution. Consequently, PhyB adheres to standard Bayesian principles, wherein the data formalizes the shift from the prior to the posterior distribution.

Based on the above theoretical results, we present an operator to approximate the evaluation of the $\eta(\pi)$.

**Definition 3** (Hybrid Belief Bellman Evaluation Operator).

The Bellman evaluation operator is defined as:

$$(\mathcal{B}^\pi Q)(s,a) = r(s,a) + \gamma \sum_{\tau_i \in \widetilde{\mathcal{T}}} \alpha_i \mathop{\mathbb{E}}_{\substack{s' \sim \tau_i \\ a' \sim \pi(\cdot|s')}} [Q(s',a')].$$

(6)

**Theorem 1.** *The Hybrid Belief Bellman Evaluation Operator $\mathcal{B}^\pi$ (Eq. (6)) is a $\gamma$-contraction. Repeatedly applying the operator $\mathcal{B}^\pi$ to any initial function $Q : \mathcal{S} \times \mathcal{A} \to \mathbb{R}$ generates a sequence that converges to $Q^\pi$. With probability at least 1-$\delta$, the objective $\eta(\pi)$ (Eq. (3)) and $Q^\pi$ satisfy:*

$$|\mathbb{E}_{\rho_0,\pi}[Q^\pi] - \eta(\pi)| \leq \frac{2\gamma R_{\max}}{(1-\gamma)^2} \sqrt{\frac{\sum_{i=0}^{k-1} \alpha_i^2}{2} \ln\left(\frac{2|S||A|}{\delta}\right)}.$$

**Remark.** Although the Hybrid Belief Bellman Evaluation Operator (Eq. (6)) is a practical approximation of the theoretical Bellman evaluation operator (Eq. (4)), Theorem 1 guarantees that the discrepancy between their fixed points is bounded. The operator $\mathcal{B}^\pi$ can be interpreted as evaluating a weighted average over a pessimistic subset $\widetilde{\mathcal{T}} \subseteq \mathcal{T}$, allowing optimization of an arbitrary $\kappa$-quantile at each time step, where $\kappa \in [0, \frac{k-1}{N-1}]$.

### 3.2. Theoretical Analysis

We investigate two theoretical questions. First, we quantify the gap between $\eta(\pi)$ (Eq. (3)) and the true performance $\eta(\pi, \tau)$ (Eq. (1)). Second, we analyze the monotonicity of $\eta(\pi)$ with respect to $N$ and $k$. All proofs are deferred to Appendix A.

**Theorem 2** (Pessimism). *Let the event $\mathcal{E} \triangleq \{\tau \in \mathcal{T}\}$ occur with probability at least 1-$\delta$. Then, the expected gap between $\eta(\pi)$ and $\eta(\pi, \tau)$ satisfies:*

$$\mathbb{E}_{\mathcal{T}}\left[\eta(\pi) - \eta(\pi, \tau)\right] \leq \frac{2\delta\gamma^2 R_{\max}}{(1-\gamma)^3} d_{\max} + \frac{\lambda\gamma \log k}{1-\gamma},$$

*where $d_{\max} = \sup_{s.t. \neg\mathcal{E}} d_{\mathrm{TV}}(\tau, \tau_{proj})$ denotes the supremum of the total variation distance between $\tau$ and its projection onto the pessimistic subset $\widetilde{\mathcal{T}}$.*

**Remark.** In practice, the condition that the event $\mathcal{E} \triangleq \{\tau \in \mathcal{T}\}$ occurs with probability at least $1 - \delta$ is readily satisfied under typical experimental configurations. First, given sufficient data coverage, training a dynamics model with high predictive accuracy is an achievable objective. Second, consistent with established literature (Guo et al., 2022; Ni et al., 2026), employing a large initial model pool is critical, from which the ensemble $\mathcal{T}$ is constructed via i.i.d. sampling $N$ times.

**Remark** (Lower Bound). Theorem 2 establishes that $\eta(\pi)$ constitutes a strict lower bound for the true expected return $\eta(\pi, \tau)$, provided that event $\mathcal{E}$ holds almost surely. Since the model ensemble $\mathcal{T}$ is generated through prior sampling, the belief $\mathbb{P}(\tau)$ must incorporate prior knowledge regarding the environment dynamics. Furthermore, Theorem 2 demonstrates that the gap between $\eta(\pi)$ and $\eta(\pi, \tau)$ vanishes as

model fidelity increases and the size of the pessimistic subset $\widetilde{\mathcal{T}}$ decreases.

**Theorem 3** (Monotonicity of Pessimism). *Let $Q^\pi$ denote the fixed point of the Hybrid Belief Bellman Evaluation Operator $\mathcal{B}^\pi$. The following properties hold:*

- *For fixed $N$, the $Q^\pi$ is monotonically non-decreasing as the size of the pessimistic subset $k$ increases.*

- *For fixed $k$, the $\mathbb{E}_{\mathcal{T}}[Q^\pi]$ is monotonically non-increasing as the ensemble size $N$ increases.*

**Remark.** Theorem 3 establishes the monotonicity of $Q^\pi$ with respect to the pessimistic subset size $k$ and model ensemble size $N$. According to Theorem 1, it follows that $\eta(\pi)$ is also monotonic in $k$ and $N$. A larger $k$ relaxes conservatism by ensuring a non-decreasing fixed point. Although the logarithmic bound in Theorem 2 loosens as $k$ increases, it serves as a worst-case guarantee. Consequently, the inequality remains valid and confirms that the performance gap is always bounded.

**Remark** (Connection to Thompson Sampling). Our approach can be conceptually framed as a pessimistic variant of Thompson Sampling (TS). While standard TS leverages the variance of the posterior distribution to encourage exploration (optimism in the face of uncertainty), our method constructs a posterior $\widetilde{\mathbb{P}}(\tau)$ that biases probability mass towards models with lower value estimates. Consequently, our sampling procedure explicitly implements pessimism in the face of uncertainty. Unlike standard robust RL that relies on a worst-case point estimate, our method preserves the stochastic nature of TS and maintains the multimodal geometry rather than collapsing to a single Dirac point.

## 4. Regularized Policy Optimization with PhyB

We begin by analyzing the challenges in optimizing the objective function, then develop a policy iteration algorithm and provide an implementation.

### 4.1. Challenges in Policy Optimization with PhyB

Optimizing $\eta(\pi)$ (Eq. (3)) presents two key challenges:

The multimodal predictive distribution induced by our posterior belief $\widetilde{\mathbb{P}}(\tau)$ poses the first optimization challenge that precludes standard algorithms operating in Euclidean space. Prior methods are mathematically equivalent to minimizing an expected squared $L_2$ distance, $\min_\theta \mathbb{E}_{\tau \sim \widetilde{\mathbb{P}}}[\|\theta - \tau\|_2^2]$. The unique solution to this objective is the arithmetic mean $\theta^* = \mathbb{E}[\tau]$. For a multimodal distribution, this solution serves as a statistically unrepresentative summary. It often resides in a region of low probability density, such as the area between two distinct modes. Consequently, policy optimization under the mean model $\theta^*$ leads to suboptimal

**Algorithm 1** Regularized policy optimization with PhyB.

**Require:** Dataset $\mathcal{D}$, model pool size $M$, model ensemble size $N$, pessimistic subset size $k$, iteration steps $G$, potential function $\psi$.

1: **Initialization:** Randomly initialize Q-function $Q_\theta(s, a)$ and policy $\pi_\phi(a|s)$. Randomly initialize target Q-function $Q_{\theta'}(s, a)$ and reference policy $\mu_{\phi'}(a|s)$ with $\theta' \leftarrow \theta$, $\phi' \leftarrow \phi$. Randomly initialize $M$ dynamics models $\{\tau_{\nu_i}(s'|s, a)\}_{i=1}^M$, forming the model pool.

2: **Dynamics model training:** Train each dynamics model $\tau_{\nu_i}(s'|s, a)$ to maximize:

$$\mathop{\mathbb{E}}_{(s_t, a_t, r_{t+1}, s_{t+1}) \sim \mathcal{D}} [\log \tau_{\nu_i}(s_{t+1}, r_{t+1}|s_t, a_t)].$$

3: **for** $i = 1, 2, \cdots, G$ **do**

4:     **Constructing $\mathcal{T}$:** Construct the model ensemble $\mathcal{T}$ by drawing $N$ independent samples from the model pool according to the prior distribution $\mathbb{P}(\tau)$.

5:     **Constructing $\widetilde{\mathcal{T}}$:** The pessimistic subset $\widetilde{\mathcal{T}}$ is formed by selecting the bottom-$k$ models from the ensemble $\mathcal{T}$, ranked by their $q_{\tau_i}(s, a)$ values.

6:     **Constructing objective function:** Construct generalized distance $D_\psi$ from potential function $\psi$, then define objective function $\widehat{\eta}(\pi_\phi, \pi_{\phi'})$.

7:     **Policy evaluation:** Perform evaluation according to Eq. (9), and update $Q_\theta$ to approximate the optimal Q-function.

8:     **Policy improvement:** Perform improvement according to Eq. (10) to update policy $\pi_\phi(a|s)$.

9:     **Moving average:** Update target Q-function and reference policy $\mu$.

10: **end for**

11: **Return:** $\pi_\phi$.

performance. This limitation motivates an optimization algorithm that respects the non-Euclidean geometry of the probability distribution space.

The second challenge arises from the evaluation of the objective $\eta(\pi)$. This process requires a complete inner dynamic programming procedure to solve the Bellman fixed-point equation $Q^\pi = \mathcal{B}^\pi Q^\pi$ (Theorem 1). However, this computationally intensive process yields only a single gradient $\nabla_\pi \eta(\pi)$ for policy improvement, making the overall optimization inefficient.

To address the first challenge, we argue that gradient-based methods should account for the geometric structure of the distribution space induced by our PhyB. To address the second challenge, we decompose the optimization into a sequence of subproblems regularized by Bregman divergence. We then design an optimal Bellman operator and prove that solving each subproblem monotonically improves

the objective $\eta(\pi)$ (Eq. (3)).

## 4.2. Bregman-Regularized Policy Iteration

Assume potential function $\psi(x)$ is strictly convex, we then define the generalized distance based on Bregman divergence:

$$D_\psi(x, y) = \psi(x) - \psi(y) - \langle \nabla \psi(y), x - y \rangle. \quad (7)$$

If $\psi(x) = \frac{1}{2}\|x\|_2^2$, then Bregman divergence reduces to the squared Euclidean distance: $D_\psi(x, y) = \frac{1}{2}\|x - y\|_2^2$. Furthermore, if $\psi(x) = \sum_{i=1}^d x_i \log x_i$ with $\sum_{i=1}^d x_i = 1$, then Bregman divergence reduces to the Kullback–Leibler (KL) divergence: $D_\psi(x, y) = \sum_{i=1}^d x_i \log \frac{x_i}{y_i}$.

**Definition 4** (Bregman-Regularized Objective Function). The Bregman-regularized objective function is defined as:

$$\widehat{\eta}(\pi, \mu) = \mathop{\mathbb{E}}_{\substack{\rho_{0,\pi} \\ \tilde{\tau}_0 \sim \widetilde{\mathbb{P}}(\tau)}} \left[ \mathop{\mathbb{E}}_{\substack{\tilde{\tau}_0, \pi \\ \tilde{\tau}_1 \sim \widetilde{\mathbb{P}}(\tau)}} \left[ \cdots \mathop{\mathbb{E}}_{\tilde{\tau}_\infty, \pi} \left[ \widehat{\mathcal{R}}_{s,a} \right] \right] \right], \quad (8)$$

where $\mu$ denotes the reference policy and $\widehat{\mathcal{R}}_{s,a} = \sum_{t=0}^\infty \gamma^t (r(s_t, a_t) - \beta D_\psi(\pi(\cdot|s), \mu(\cdot|s)))$.

**Definition 5** (Optimal Hybrid Belief Bellman Operator). To optimize $\widehat{\eta}(\pi, \mu)$, we define the Optimal Hybrid Belief Bellman Operator $\widehat{\mathcal{B}}^*$ based on pessimistic subset $\widetilde{\mathcal{T}}$:

$$(\widehat{\mathcal{B}}^* Q)(s, a) = r(s, a) + \gamma \sum_{\tau_i \in \widetilde{\mathcal{T}}} \alpha_i \mathop{\mathbb{E}}_{s' \sim \tau_i} [V^*(s')], \quad (9)$$

where the regularized value function $V^*(s)$ is given by:

$$V^*(s) = \max_\pi \left\{ \mathop{\mathbb{E}}_{a \sim \pi} [Q(s, a)] - \beta D_\psi(\pi(\cdot|s), \mu(\cdot|s)) \right\}.$$

Following a derivation similar to that in Theorem 1, the optimal hybrid belief Bellman operator provides a tractable approximation to $\widehat{\eta}(\pi, \mu)$. We now examine its key properties, as formalized in the following theorem:

**Theorem 4.** *The Optimal Hybrid Belief Bellman Operator $\widehat{\mathcal{B}}^*$ (Eq.(9)) is a $\gamma$-contraction. Repeatedly applying the operator $\widehat{\mathcal{B}}^*$ to any initial function $Q : \mathcal{S} \times \mathcal{A} \to \mathbb{R}$ leads to a sequence converging to $Q^*$. The corresponding optimal policy is*

$$\pi^* = (\nabla \psi)^{-1} \left( \frac{1}{\beta}(Q^* - z) + \nabla \psi(\mu) \right), \quad (10)$$

*where $z$ denotes the Lagrange multiplier introduced to enforce probability normalization.*

Since setting a task-specific reference policy is challenge, we adopt an iterative approach. During the $i$-th iteration, we set $\mu$ to the optimal policy $\pi_{i-1}$ obtained from the $(i-1)$-th iteration. Although the regularized objective $\widehat{\eta}(\pi_i, \pi_{i-1})$ differs from the true objective $\eta(\pi_i)$ (Eq. (3)) , we show that optimizing the former provably improves the latter.

*Table 1.* D4RL Results. Normalized scores are computed as $100 \times$ (score - random policy score) / (expert policy score - random policy score), reported as mean ± standard deviation. The score of our proposed approach is averaged over 4 random seeds.

| | | Model-free methods | | | | | Model-based methods | | | | |
|---|---|---|---|---|---|---|---|---|---|---|---|
| | | EPQ | CQL | FQL | TD3+BC | DMG | MOReL | RAMBO | PMDB | ADM | (Ours) PhyB |
| Random | HalfCheetah | 33.0±2.4 | 31.3±3.5 | 14.2±0.5 | 10.2±1.3 | 28.8±1.3 | 38.9±1.8 | 39.5±3.5 | 37.8±0.2 | **45.4±2.8** | 34.7±1.7 |
| | Hopper | 32.1±0.3 | 5.3±0.6 | 9.6±0.2 | 11.0±0.1 | 20.4±10.4 | **38.1±10.1** | 25.4±7.5 | 32.7±0.1 | 32.7±0.2 | 33.9±1.1 |
| | Walker2d | 23.0±0.7 | 5.4±1.7 | 4.1±0.1 | 1.4±1.6 | 4.8±2.2 | 16.0±7.7 | 0.0±0.3 | 21.8±0.1 | 22.2±0.2 | **23.5±1.5** |
| Medium | HalfCheetah | 67.3±0.5 | 46.9±0.4 | 59.9±0.5 | 42.8±0.3 | 54.9±0.2 | 60.7±4.4 | **77.9±4.0** | 75.6±1.3 | 72.2±0.6 | 74.5±1.8 |
| | Hopper | 101.3±0.2 | 61.9±6.4 | 44.3±3.1 | 99.5±1.0 | 100.6±1.9 | 84.0±17.0 | 87.0±15.4 | 106.8±0.2 | 107.4±0.6 | **109.4±2.0** |
| | Walker2d | 87.8±2.1 | 79.5±3.2 | 9.5±1.8 | 79.7±1.8 | 92.4±2.7 | 72.8±11.9 | 84.9±2.6 | 94.2±1.1 | 93.2±1.1 | **95.5±8.5** |
| Expert | HalfCheetah | 107.2±0.2 | 97.3±1.1 | 4.4±1.2 | 105.7±1.9 | 95.9±0.3 | 8.4±11.8 | 79.3±15.1 | 105.7±1.0 | 89.4±26.4 | **113.7±1.0** |
| | Hopper | 112.4±0.5 | 106.5±9.1 | 40.5±6.8 | 112.2±0.2 | 111.5±2.2 | 80.4±34.9 | 50.0±8.1 | 111.7±0.3 | 102.3±11.9 | **118.9±2.3** |
| | Walker2d | 109.8±1.0 | 109.3±0.1 | 12.7±2.7 | 105.7±2.7 | 114.7±0.4 | 62.6±29.9 | 1.6±2.3 | 115.9±1.9 | 5.5±1.3 | **116.3±1.1** |
| Medium Expert | HalfCheetah | 95.7±0.3 | 95.0±1.4 | 97.5±9.4 | 97.9±4.4 | 91.1±4.2 | 80.4±11.7 | 95.4±5.4 | 108.5±0.5 | 103.7±0.2 | **109.4±1.5** |
| | Hopper | 108.8±5.2 | 96.9±15.1 | 43.8±5.6 | 112.2±0.2 | 110.4±3.4 | 105.6±8.2 | 88.2±20.5 | 111.8±0.6 | 112.7±0.3 | **116.5±2.1** |
| | Walker2d | 112.0±0.6 | 109.1±0.2 | 7.3±1.2 | 101.1±9.3 | 114.4±0.7 | 107.5±5.6 | 56.7±39.0 | 111.9±0.2 | **114.9±0.3** | 112.4±1.1 |
| Medium Replay | HalfCheetah | 62.0±1.6 | 45.3±0.3 | 52.2±0.5 | 43.3±0.5 | 51.4±0.3 | 44.5±5.6 | 68.7±5.3 | 71.7±1.1 | 67.6±3.4 | **74.7±1.5** |
| | Hopper | 97.8±1.0 | 86.3±7.3 | 40.6±3.6 | 31.4±3.0 | 101.9±1.4 | 81.8±17.0 | 99.5±4.8 | 106.2±0.6 | 104.4±0.4 | **110.7±1.3** |
| | Walker2d | 85.3±1.0 | 76.8±10.0 | 11.4±3.5 | 25.2±5.1 | 89.7±5.0 | 40.8±20.4 | 89.2±6.7 | 79.9±0.2 | **95.6±2.1** | 85.4±3.9 |
| Full Replay | HalfCheetah | 85.3±0.7 | 76.9±0.9 | 80.9±0.5 | 71.9±2.7 | 79.9±1.2 | 70.1±5.1 | 87.0±3.2 | 90.0±0.8 | 86.3±1.7 | **98.1±3.5** |
| | Hopper | 108.5±0.6 | 101.9±0.6 | 89.6±3.6 | 85.9±16.4 | 106.4±1.1 | 94.4±20.5 | 105.2±2.1 | 109.1±0.2 | 108.5±0.7 | **111.0±1.0** |
| | Walker2d | **107.4±0.6** | 94.2±1.9 | 42.2±13.1 | 92.0±3.6 | 97.5±4.6 | 84.8±13.1 | 88.3±4.9 | 95.4±0.7 | 99.9±3.6 | 99.7±0.9 |
| Average | | 85.4 | 73.7 | 36.9 | 68.3 | 81.5 | 65.1 | 68.0 | 88.2 | 81.3 | **91.0** |

*Table 2.* Normalized scores on offline optimal liquidation task.

| Methods | PhyB (Ours) | PMDB | 1R2R | ORAAC | RAMBO | CQL | IQL | TD3+BC | MOPO | COMBO |
|---|---|---|---|---|---|---|---|---|---|---|
| Score | **101.6±3.7** | 85.5±2.3 | 78.8±1.6 | 0.0±0.0 | 99.6±0.6 | 89.4±1.3 | 0.0±0.0 | 100.4±3.6 | 64.7±21.8 | 55.0±29.0 |

**Theorem 5** (Monotonic Improvement). *Starting from an arbitrary initial policy $\pi_0$, consider the sequence of policies $\{\pi_i\}$ generated by iteratively solving the Bregman-regularized subproblem: $\pi_{i+1} = (\nabla\psi)^{-1}\big(\frac{1}{\beta}(Q - z) + \nabla\psi(\pi_i)\big)$. Then we have: $\eta(\pi_{i+1}) \geq \eta(\pi_i)$.*

**Remark** (Convergence Analysis). The policy sequence $\{\pi_i\}_{i \geq 0}$ improves monotonically with respect to the objective function $\eta(\pi)$ defined in Eq. (3). Together with Theorem 4, the sequence will converge to the optimal policy $\pi^* = (\nabla\psi)^{-1}\big(\frac{1}{\beta}(Q^* - z) + \nabla\psi(\mu)\big)$.

### 4.3. Algorithmic Implementation

The pseudocode is presented in Algorithm 1. The Q-function $Q_\theta$ and policy $\pi_\phi$ are parameterized by neural networks. Transition dynamics are modeled using a Gaussian distribution over next states and rewards (Yu et al., 2020; Kidambi et al., 2020; Guo et al., 2022).

The prior belief $\mathbb{P}(\tau)$ over dynamics models provides a mechanism to incorporate prior knowledge about the environment. However, real-world scenarios often lack complete expert knowledge. An effective alternative is to employ a uniform prior over an ensemble of learned dynamics models (Chua et al., 2018), which provides a standard baseline by

naturally assigning lower confidence to OOD regions. In our implementation, to computationally approximate this continuous sampling process, we establish a large candidate model pool of size $M$ (where $M \gg N$). The model ensemble $\mathcal{T}$ is then constructed by drawing $N$ independent samples from this pool according to the prior distribution. In practice, PhyB's inference remains cheap, but its training time scales linearly with the ensemble size $N$, as computing $\{q_{\tau_i}(s, a)\}_{i=0}^{N-1}$ requires a forward pass through all $N$ models. This bottleneck can be mitigated by distributed training across multiple GPUs.

We adopt the potential function $\psi(x) = \frac{1-\omega}{2}\|x\|_2^2 + \omega \sum_{i=1}^d x_i \log x_i$ (Huang et al., 2022). Due to its twice differentiability, the gradient update in the policy improvement step can be approximated by the following expression:

$$\phi_{t+1} = \phi_t + \frac{1}{\beta}[\nabla^2\psi(\phi_t)]^{-1}g_t. \tag{11}$$

This formulation mirrors the natural gradient method (Kakade, 2001), with $\nabla^2\psi(\phi_t)$ serving as the Riemannian metric on a composite manifold of information-geometric and Euclidean structures. Since directly evaluating the inverse Hessian $[\nabla^2\psi(\phi_t)]^{-1}$ is computationally intractable in deep RL settings, we instead implement the policy up-

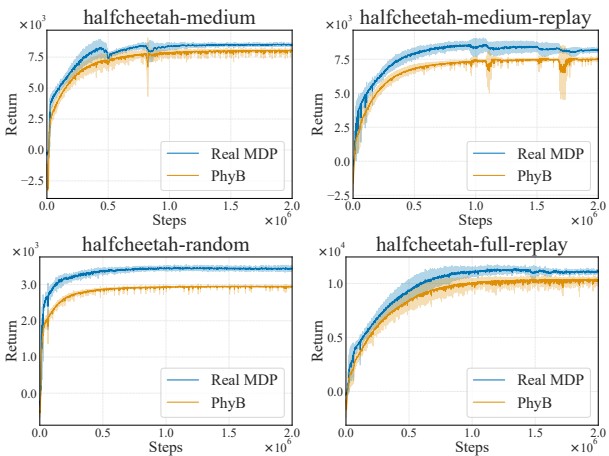

*Figure 1.* Learning and evaluation curves in HalfCheetah-v2 environment.

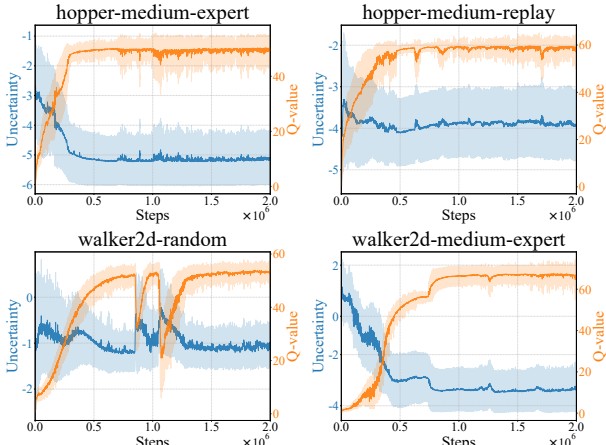

*Figure 2.* Evolution of Q-values and uncertainty for encountered state-action pairs during training. For a given $(s, a)$, uncertainty is quantified as: $\log\left(\mathrm{std}(\mathbb{E}_{s' \sim \tau}[s'])\right), \tau \in \widetilde{\mathcal{T}}$.

*Table 3.* Impact of model ensemble size $N$ and pessimistic subset size $k$ on policy performance.

| $(N, k)$ | Hopper-M-E | Walker2d-M-E | HalfCheetah-M-E |
|---|---|---|---|
| (5,5) | 119.9±1.5 | 118.0±1.4 | 113.9±0.7 |
| (6,5) | 118.7±0.9 | 113.9±0.3 | 110.1±1.2 |
| (8,5) | 118.1±1.1 | 114.7±0.7 | 112.2±1.7 |
| (12,5) | 112.6±2.2 | 108.2±1.5 | 106.4±2.3 |
| (10,4) | 112.8±1.5 | 110.6±1.2 | 95.5±2.8 |
| (10,6) | 116.8±1.0 | 113.5±0.7 | 109.7±0.7 |
| (10,8) | 119.4±1.4 | 115.0±1.0 | 111.2±1.1 |

*Table 4.* Sensitivity analysis of the hyperparameter $\omega$. "M" denotes the "medium" dataset type.

| $\omega$ | Hopper-M | Walker2d-M | HalfCheetah-M |
|---|---|---|---|
| 0.0 | 104.3±3.6 | 92.0±3.1 | 71.9±3.1 |
| 0.1 | 105.6±4.9 | 92.8±4.2 | 71.7±2.1 |
| 0.3 | 107.0±1.6 | 93.5±0.6 | 72.5±1.6 |
| 0.5 | 112.2±1.2 | 98.2±0.8 | 73.4±1.1 |
| 0.7 | 107.7±0.9 | 86.4±0.8 | 71.9±1.2 |
| 0.9 | 109.4±2.0 | 95.5±8.5 | 74.5±1.8 |
| 1.0 | 104.6±6.1 | 92.3±8.7 | 70.3±3.5 |

date via an equivalent mirror descent formulation (see Appendix C) to maintain scalability.

## 5. Experiments

Our experiments address three research questions (RQs).

**Performance (RQ1):** How does PhyB compare to previous SoTA offline RL algorithms on standard benchmarks?

**Empirical Validation (RQ2):** How well do the experimental results align with the theoretical findings when neural networks are used as function approximators?

**Ablation Study (RQ3):** How does each design component of PhyB contribute to overall performance?

To answer the above questions, we conduct experiments on the D4RL benchmark (Fu et al., 2020) and a stochastic offline optimal liquidation benchmark (Bao & Liu, 2019; Rigter et al., 2023). Detailed hyperparameters and experimental configurations are provided in Appendix D. This appendix also comprehensively documents supplementary results, such as computational overhead benchmarks and sensitivity analyzes regarding the model pool size.

### 5.1. Performance (RQ1)

**Results on D4RL benchmarks.** In response to RQ1, we evaluate PhyB against a range of SoTA methods, including model-free algorithms: EPQ (Lin et al., 2024), CQL (Kumar et al., 2020), FQL (Park et al., 2025), TD3+BC (Fujimoto & Gu, 2021), DMG (Mao et al., 2024), and model-based approaches: MOReL (Kidambi et al., 2020), RAMBO (Rigter et al., 2022), PMDB (Guo et al., 2022), ADM (Lin et al., 2025). As detailed in Table 1, PhyB achieves superior performance on 12 datasets and remains competitive on the remaining 6. While reference results for the algorithms in the table are drawn from their original publications and (Guo et al., 2022), we reproduce any results not provided in those works (such as ADM, DMG,TD3+BC and FQL) by implementing the hyperparameter configurations from the original papers. The results reveal that PhyB's performance improves substantially with the inclusion of even a limited amount of high-quality data, as evidenced by its strong performance on the "medium," "medium-expert," and "full-replay" datasets. We attribute this improvement to PhyB's multimodal posterior belief, which effectively leverages the generalization capacity of the dynamics models trained on these datasets. See Appendix D for more results.

*Table 5.* Ablation study on the effect of the convex combination. "HC" denotes the "HalfCheetah", "Ho" denotes the "Hopper", "W2d" denotes the "Walker2d". We observe a substantial drop in performance when this component is removed, highlighting its critical role.

| Dataset | Medium | | | Expert | | | Medium Expert | | | Medium Replay | | |
|---|---|---|---|---|---|---|---|---|---|---|---|---|
| | HC | Ho | W2d | HC | Ho | W2d | HC | Ho | W2d | HC | Ho | W2d |
| Score | 68.3±2.8 | 97.6±9.4 | 81.3±11.8 | 106.9±2.2 | 109.7±6.5 | 107.5±2.5 | 97.0±3.9 | 104.3±10.1 | 99.1±3.6 | 67.5±1.6 | 97.4±7.5 | 67.1±4.3 |

*Table 6.* Ablation study on the effect of the $\lambda$. "HC" denotes the "HalfCheetah", "Ho" denotes the "Hopper", "W2d" denotes the "Walker2d". We report the percentage improvement or degradation compared to the standard $\lambda = 0.33$ setting.

| Dataset | Random | | | Medium | | | Expert | | | Medium Expert | | |
|---|---|---|---|---|---|---|---|---|---|---|---|---|
| | HC | Ho | W2d | HC | Ho | W2d | HC | Ho | W2d | HC | Ho | W2d |
| $\lambda = 0.2$ | ↑2.95% | ↓0.08% | ↓4.81% | ↑3.34% | ↓0.46% | ↑0.22% | ↓0.56% | ↓1.45% | ↓0.44% | ↓1.22% | ↑1.43% | ↓0.82% |
| $\lambda = 1.0$ | ↑0.20% | ↑0.06% | ↓5.42% | ↑0.93% | ↑0.21% | ↑2.93% | ↑2.73% | ↑1.13% | ↑4.79% | ↑2.18% | ↓0.49% | ↓0.17% |

**Results on offline optimal liquidation.** We evaluate PhyB on a custom stochastic currency liquidation task where an agent must liquidate an asset within a fixed horizon $T$ while facing a randomly fluctuating exchange rate. The high stochasticity of this environment, combined with an offline dataset generated by a random policy, presents a significant challenge to traditional offline RL methods. We defer the detailed dataset profile and experimental configurations to Appendix D.

We compare our method against SoTA model-free (ORAAC (Urpí et al., 2021), IQL (Kostrikov et al., 2022)) and model-based (MOPO (Yu et al., 2020), COMBO (Yu et al., 2021)) baselines. As shown in Table 2, these strong baselines largely fail to learn effective policies in this stochastic setting, whereas PhyB demonstrates robust and superior performance.

### 5.2. Empirical Validation (RQ2)

**Pessimism.** As shown in Figure 1, the learning curve of PhyB closely tracks and consistently lower bounds the true return. This empirical evidence supports the Theorem 2. Furthermore, the observed near-monotonic improvement validates the monotonic property established in Theorem 5.

**Uncertainty quantification.** We quantify uncertainty for $(s, a)$ via the log standard deviation of next-state predictions over the pessimistic subset $\widetilde{\mathcal{T}}$ (Figure 2). The figure shows that spikes in uncertainty consistently coincide with drops in Q-values, indicating that our method penalizes high epistemic uncertainty actions and favors in-distribution behavior. We hypothesize this uncertainty spike occurs when the policy encounters an OOD state-action pair, indicating that it has strayed beyond the support of the original dataset.

**Monotonicity.** Table 3 shows the effect of the model ensemble size $N$ and the pessimistic subset size $k$ on final policy performance. Results align with Theorem 3, show-

ing performance is non-increasing in $N$ and non-decreasing in $k$. Since the optimal $(N, k)$ configuration varies across datasets, we set $N = 10$ and $k = 5$ for a fair comparison.

### 5.3. Ablation Study (RQ3)

**Sensitivity analysis of the hyperparameter $\omega$.** We conduct a sensitivity analysis on $\omega$ (where $\omega = 1$ recovers KL-regularization). Table 4 demonstrates PhyB's robustness: while $\omega = 0.5$ minimizes the standard deviation, this metric increases gradually as $\omega$ approaches the boundaries of 0 or 1. Given the overall stability of the method, we adopt a fixed default of $\omega = 0.9$ for all experiments.

**Effect of the convex combination.** The construction of the posterior belief hinges on the coefficient $\alpha$, which is determined by solving Eq. (5). To evaluate the impact of the proposed convex combination method, we conduct an ablation study comparing it with a simplified variant that selects a single model directly from the pessimistic subset $\widetilde{\mathcal{T}}$. As shown in Table 5, the simplified variant exhibits a significant performance degradation. This result confirms the efficacy of the convex combination approach and validates the overall design of PhyB.

**Sensitivity analysis of the hyperparameter $\lambda$.** The hyperparameter $\lambda$ governs the concentration of the posterior belief: a smaller $\lambda$ shifts more weight toward highly pessimistic models, while a larger $\lambda$ leads to a more uniform distribution. Our sensitivity analysis in Table 6 reveals two insights regarding the hyperparameter $\lambda$. First, performance remains stable across a reasonable range of hyperparameter values, demonstrating that PhyB is robust to parameter variations and circumvents extensive tuning. Second, we identify a trade-off governed by data quality: lower-quality datasets benefit from a smaller $\lambda$ to enforce higher pessimism against OOD risks, whereas in high-quality domains, an overly small $\lambda$ induces excessive conservatism that leads to performance decay. This observation confirms that $\lambda$

effectively balances empirical reward maximization with robust dynamics regularization based on data reliability.

# 6. Related Works

## 6.1. Uncertainty in Model-Based Offline RL

Model-based offline RL trains a dynamics model on an offline dataset and then uses synthetic rollouts from this model to optimize policies (Luo et al., 2024). However, due to inevitable model errors, directly optimizing policies using the learned model often leads to substantial performance degradation (Kidambi et al., 2020). Existing methods typically apply pointwise penalties based on epistemic uncertainty quantification (e.g., the standard deviation of ensemble predictions) to prevent the policy from exploring OOD regions (Yu et al., 2020; Sun et al., 2023). Prediction error in OOD regions is typically highly correlated with epistemic uncertainty (Guo et al., 2022). This relationship motivates several methods to explicitly penalize Q-value estimates for synthetic OOD samples generated by the dynamics model (Yu et al., 2021; Park & Lee, 2025). As penalty designs rely on epistemic uncertainty, inaccurate quantification often induces performance degradation and weak generalization (Park et al., 2024; Qiao et al., 2025). Given that increased data coverage naturally diminishes epistemic uncertainty, recent works focus on generating diverse or high-fidelity trajectories to achieve more robust uncertainty estimation (Zhai et al., 2024; Lin et al., 2025).

While existing research focus on quantifying sample-level epistemic uncertainty, they overlook the uncertainty of the transition dynamics. Incorporating the uncertainty of the transition dynamics and analyzing its effect on long-term cumulative reward is non-trivial in the standard RL framework, as the dynamics model is learned independently of the value-maximization objective. We address this decoupling by treating the dynamics model as a random variable rather than a deterministic point estimate. By integrating this uncertainty directly into the optimization objective, our approach ensures that the learned value function is robust to the uncertainty of the transition dynamics.

## 6.2. Bayesian RL

Bayesian RL treats transition dynamics as random variables rather than deterministic point estimates (Lin et al., 2026a). This probabilistic formulation not only offers a formal framework to quantify epistemic uncertainty (Derman et al., 2020; Wei et al., 2023), but also emerges as a pessimism-free paradigm in the offline RL (Ni et al., 2026; Lin et al., 2026b). Despite its theoretical advantages, policy optimization in Bayesian RL involves a composite objective over an infinite-dimensional function space (Jiang et al., 2025), rendering posterior inference intractable. Early

works address this objective function using mixed-integer linear programming (Lobo et al., 2020) or Monte Carlo tree search (Rigter et al., 2021). However, these methods remain computationally intractable and scale poorly to high-dimensional environments. To improve computational tractability, recent works adopt robust MDP formulations to account for worst-case transitions (Rigter et al., 2022; Dong et al., 2024). Although robust MDPs effectively mitigate model exploitation (Ghosh et al., 2022), they implicitly rely on Dirac likelihoods that concentrate probability mass on the most adversarial dynamics, leading to excessive pessimism (Guo et al., 2022). Subsequent research addresses the scalability issue by assuming a fixed posterior distribution derived from offline datasets (Rigter et al., 2023). While these simplifications facilitate practical implementation, they sacrifice the inherent adaptability and uncertainty quantification in Bayesian RL.

Model ensembles provide diverse information that is critical for offline RL (Choi et al., 2024). Unlike prior methods, our work utilizes a multimodal likelihood to integrate all candidate models without imposing restrictive assumptions on the form of the posterior. To handle the multimodal geometry while maintaining computational efficiency, we decompose the policy optimization task into a sequence of subproblems. We then develop a Bregman-regularized iterative algorithm to solve these subproblems, ensuring monotonic policy improvement until convergence.

# 7. Discussion and Conclusion

In this work, we present the Posterior Hybrid Bayesian Belief (PhyB) framework along with a regularized iterative algorithm. This approach addresses the challenge of policy optimization in Bayesian RL without relying on restrictive assumptions. Our experimental results demonstrate that PhyB consistently outperforms prior methodologies across diverse benchmarks. Furthermore, the empirical findings align with our theoretical analysis, confirming the effectiveness of the proposed method. Conceptually, PhyB aligns with risk-averse RL through its treatment of uncertainty. By maintaining a distribution over a model ensemble, PhyB formally quantifies epistemic uncertainty in a unified manner. Selecting the $k$ lowest model-predicted value estimates operates as a pessimistic pruning mechanism, directly mirroring Conditional Value-at-Risk (CVaR) which optimizes for the lower tail of a probability distribution. Unlike heuristic reward penalties that rely on manually tuned coefficients, this implicit CVaR mechanism inherently adapts to the degree of model disagreement. In OOD regions where predictions diverge, the lower-tail evaluation automatically intensifies to suppress value overestimation. This integration of uncertainty quantification and tail-risk optimization provides a framework for stable policy update under distribution shifts.

## Acknowledgment

We thank the anonymous reviewers for their valuable feedback on an early version of this paper. This work was supported by the Major Program of the National Natural Science Foundation of China (No. T2293723) and the Young Scientists Fund of the National Natural Science Foundation of China (No. 62506335).

## Impact Statement

This paper presents work whose goal is to advance the field of Machine Learning. There are many potential societal consequences of our work, none of which we feel must be specifically highlighted here.

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

# A. Proofs

**Definition 1** (Pessimistic Subset). Given a finite model ensemble $\mathcal{T} = \{\tau_0, \ldots, \tau_{N-1}\}$ of size $N$, where each $\tau_i$ is sampled i.i.d. from the prior $\mathbb{P}(\tau)$, the pessimistic subset $\widetilde{\mathcal{T}} \subseteq \mathcal{T}$ is formed by selecting the $k$ models that correspond to the bottom-$k$ $q_{\tau_i}(s, a) = \mathbb{E}_{\tau_i, \pi}[Q(s', a')]$ values.

**Proposition 1** (Likelihood Ratio). *Let $N$ and $k$ denote the sizes of the model ensemble $\mathcal{T}$ and pessimistic subset $\widetilde{\mathcal{T}}$, respectively. Let $\mathcal{F}$ denote the cumulative distribution function of the $q_{\tau_i}(s, a)$. Then the likelihood is given by:*

$$\xi(q_\tau) = \sum_{i=0}^{k-1} \alpha_i \frac{N!}{i!(N-i-1)!} [\mathcal{F}(q_\tau)]^i [1 - \mathcal{F}(q_\tau)]^{N-i-1},$$

*where the weights $\tilde{\alpha} = \{\alpha_i | \sum_{i=0}^{k-1} \alpha_i = 1, \alpha_i \geq 0\}$ are the solution to the following entropy-regularized problem:*

$$\min_{\tilde{\alpha}} \sum_{i=0}^{k-1} \alpha_i q_{\tau_i}(s, a) + \lambda \alpha_i \log \alpha_i.$$

*Proof of Proposition 1.* Let $q_{(0)} \leq q_{(1)} \leq \cdots \leq q_{(N-1)}$ denote the order statistics of the scalar Q-values $\{q_{\tau_i}(s, a)\}_{i=0}^{N-1}$ generated by the ensemble. The probability density function (PDF) of the $i$-th order statistic at value $q$ is given by:

$$f_i(q) = \frac{N!}{i!(N-i-1)!} [\mathcal{F}(q)]^i [1 - \mathcal{F}(q)]^{N-i-1} f(q),$$

where $f(q)$ and $\mathcal{F}(q)$ are the PDF and CDF of the prior belief, respectively.

Our goal is to identify a weighting function $\xi(q)$ such that the weighted sum of expected order statistics equals the expectation under a twisted belief $\widetilde{\mathbb{P}}$. We begin our analysis with the LHS (the pessimistic objective):

$$\text{LHS} = \mathbb{E}\left[\sum_{i=0}^{k-1} \alpha_i q_{(i)}\right] = \sum_{i=0}^{k-1} \alpha_i \int_{-\infty}^{\infty} q \cdot f_i(q)\, dq.$$

Substituting the definition of $f_i(q)$ into the equation:

$$\text{LHS} = \int_{-\infty}^{\infty} q \cdot \left(\sum_{i=0}^{k-1} \alpha_i \frac{N!}{i!(N-i-1)!} [\mathcal{F}(q)]^i [1 - \mathcal{F}(q)]^{N-i-1}\right) f(q)\, dq.$$

Regarding the RHS, we formulate the new belief $\widetilde{\mathbb{P}}$ through a likelihood ratio $\xi(q)$ acting on the value space. Let $\Phi : \tau \mapsto q_\tau$, then:

$$\begin{aligned}
\text{RHS} = \mathbb{E}_{\tau \sim \widetilde{\mathbb{P}}}[q_\tau] &= \int q_\tau\, d\widetilde{\mathbb{P}}(\tau) \\
&= \int q_\tau \cdot \frac{d\widetilde{\mathbb{P}}}{d\mathbb{P}}(\tau)\, d\mathbb{P}(\tau) \\
&= \int \Phi(\tau) \cdot \xi_N(\Phi(\tau))\, d\mathbb{P}(\tau) \\
&= \int q \cdot \xi_N(q) f(q)\, dq.
\end{aligned}$$

By matching the integral forms of LHS and RHS, we identify the explicit form of the likelihood ratio:

$$\xi_N(q) = \sum_{i=0}^{k-1} \alpha_i \frac{N!}{i!(N-i-1)!} [\mathcal{F}(q)]^i [1 - \mathcal{F}(q)]^{N-i-1}.$$

Specifically, we define the Radon-Nikodym derivative through the evaluation map $\Phi : \tau \mapsto q_\tau$:

$$\frac{d\widetilde{\mathbb{P}}}{d\mathbb{P}}(\tau) := (\xi_N \circ \Phi)(\tau) = \xi_N(\Phi(\tau)) = \xi_N(q_\tau).$$

To analyze the property of the weights, let $\Xi(u) = u^i(1-u)^{N-i-1}$ with $u = \mathcal{F}(q)$. The derivative with respect to $u$ is:

$$\frac{d\Xi}{du} = u^{i-1}(1-u)^{N-i-2}(i - u(N-1)).$$

We observe that $\frac{d\Xi}{du} \geq 0$ for $u \leq \frac{i}{N-1}$ and $\frac{d\Xi}{du} \leq 0$ otherwise. Thus, the weighting term achieves its maximum when the CDF value is $\frac{i}{N-1}$. This corresponds to the Q-value $q_{\tau^*}$ being the $\frac{i}{N-1}$-quantile of the prior distribution. $\square$

**Theorem 1.** *Hybrid Belief Bellman Evaluation Operator $\mathcal{B}^\pi$ is a contraction mapping. Repeatedly applying the operator $\mathcal{B}^\pi$ to any initial function $Q : \mathcal{S} \times \mathcal{A} \to \mathbb{R}$ generates a sequence that converges to $Q^\pi$. With probability at least 1-$\delta$, the objective function $\eta(\pi)$ and $Q^\pi$ satisfy: $|\mathbb{E}_{\rho_0,\pi}[Q^\pi] - \eta(\pi)| \leq \frac{2\gamma R_{\max}}{(1-\gamma)^2}\sqrt{\frac{\sum_{i=0}^{k-1}\alpha_i^2}{2}\ln\left(\frac{2|S||A|}{\delta}\right)}.$*

*Proof of Proposition 1.* The Bellman operator is defined as:

$$\mathcal{B}^\pi Q(s,a) = r(s,a) + \gamma \sum_{\tau_i \in \widetilde{\mathcal{T}}} \alpha_i \mathop{\mathbb{E}}_{\substack{s' \sim \tau_i \\ a' \sim \pi(\cdot|s')}} [Q(s',a')].$$

Based on the result of Proposition 1 and statistical knowledge, we can obtain the following relation:

$$\sum_{\tau_i \in \widetilde{\mathcal{T}}} \alpha_i \mathop{\mathbb{E}}_{s' \sim \tau_i, a' \sim \pi(\cdot|s')} [Q(s',a')] \xrightarrow{|\mathcal{T}| \to \infty} \mathop{\mathbb{E}}_{\tau \sim \widetilde{\mathbb{P}}(\tau)} \left[ \mathop{\mathbb{E}}_{s' \sim \tau(s,a), a' \sim \pi(\cdot|s')} [Q(s',a')] \right].$$

To theoretically guarantee that the operator $\mathcal{B}^\pi$ is a contraction, we must constrain the magnitude of $\lambda$ such that $\lambda > \frac{\gamma \cdot \Delta_{\max}}{2(1-\gamma)}$, where $\Delta_{\max} = \sup_{s,a}(\max_i q_{\tau_i}(s,a) - \min_i q_{\tau_i}(s,a))$.

Let $f(\mathbf{q}) = \sum_{i=0}^{k-1} \alpha_i(\mathbf{q})q_i$, where $\alpha_i(\mathbf{q}) = \frac{\exp(-q_i/\lambda)}{\sum_j \exp(-q_j/\lambda)}$ and $q_i$ denotes $q_{\tau_i}$. Then we have:

$$\frac{\partial f}{\partial q_j} = \alpha_j - \frac{1}{\lambda}\sum_{i=0}^{k-1} q_i \alpha_i(\mathbb{I}(i=j) - \alpha_j)$$

$$= \alpha_j - \frac{1}{\lambda}\left(q_j\alpha_j - \alpha_j\sum_{i=0}^{k-1}\alpha_i q_i\right)$$

$$= \alpha_j\left(1 - \frac{1}{\lambda}(q_j - \mathbb{E}_\alpha[\mathbf{q}])\right),$$

where $\mathbb{E}_\alpha[\mathbf{q}] = f(\mathbf{q})$. Then:

$$\|\nabla f(\mathbf{q})\|_1 \leq \sum_{j=0}^{k-1}\alpha_j\left(1 + \frac{1}{\lambda}|q_j - \mathbb{E}_\alpha[\mathbf{q}]|\right)$$

$$= 1 + \frac{1}{\lambda}\sum_{j=1}^{k}\alpha_j|q_j - \mathbb{E}_\alpha[\mathbf{q}]|$$

$$\leq 1 + \frac{\Delta_{\max}}{2\lambda}.$$

It implies that $\gamma\left(1 + \frac{\Delta_{\max}}{2\lambda}\right) < 1$.

Returning to the definition of the contractivity of the Bellman operator:

$$\|\mathcal{B}^\pi Q_1 - \mathcal{B}^\pi Q_2\|_\infty = \gamma\|f(\mathbf{q}(Q_1)) - f(\mathbf{q}(Q_2))\|_\infty$$

$$\leq \gamma \cdot \|\nabla f(\mathbf{q})\|_1 \cdot \|\mathbf{q}(Q_1) - \mathbf{q}(Q_2)\|_\infty$$

$$\leq \gamma\left(1 + \frac{\Delta_{\max}}{2\lambda}\right)\|Q_1 - Q_2\|_\infty.$$

Given that $\gamma \left(1 + \frac{\Delta_{\max}}{2\lambda}\right) < 1$, the Banach fixed-point theorem ensures that the Hybrid Belief Bellman Evaluation Operator has a unique fixed point. Based on the previous results, we can easily obtain $\mathbb{E}_{\rho_0, \pi}[Q^\pi] \xrightarrow{|\mathcal{T}| \to \infty} \eta(\pi)$.

We then define the theoretical Bellman evaluation operator $\hat{\mathcal{B}}^\pi$:

$$\hat{\mathcal{B}}^\pi Q(s,a) = r(s,a) + \gamma \underset{\substack{\tau \sim \widetilde{\mathbb{P}}(\tau) \\ s' \sim \tau, a' \sim \pi}}{\mathbb{E}} [Q(s',a')].$$

This operator is also a $\gamma$-contraction, and its unique fixed point, which we denote as $Q^{\pi^*}$. The relation between objective function $\eta(\pi)$ and $Q^{\pi^*}$ is:

$$\eta(\pi) = \mathbb{E}_{\rho_0, \pi}[Q^{\pi^*}].$$

We begin by bounding the infinity norm between $Q^{\pi^*}$ and $Q^\pi$:

$$\|Q^\pi - Q^{\pi^*}\|_\infty = \|\mathcal{B}^\pi Q^\pi - \hat{\mathcal{B}}^\pi Q^{\pi^*}\|_\infty$$
$$= \|(\mathcal{B}^\pi Q^\pi - \mathcal{B}^\pi Q^{\pi^*}) + (\mathcal{B}^\pi Q^{\pi^*} - \hat{\mathcal{B}}^\pi Q^{\pi^*})\|_\infty$$
$$\leq \|\mathcal{B}^\pi Q^\pi - \mathcal{B}^\pi Q^{\pi^*}\|_\infty + \|\mathcal{B}^\pi Q^{\pi^*} - \hat{\mathcal{B}}^\pi Q^{\pi^*}\|_\infty,$$

where the last step follows from the triangle inequality. Since $\mathcal{B}^\pi$ is a $\gamma$-contraction, we have:

$$\|\mathcal{B}^\pi Q^\pi - \mathcal{B}^\pi Q^{\pi^*}\|_\infty \leq \gamma \|Q^\pi - Q^{\pi^*}\|_\infty.$$

Then

$$(1-\gamma)\|Q^\pi - Q^{\pi^*}\|_\infty \leq \|\mathcal{B}^\pi Q^{\pi^*} - \hat{\mathcal{B}}^\pi Q^{\pi^*}\|_\infty.$$

The term on the right-hand side represents the discrepancy between the empirical operator and the theoretical operator, evaluated at the $Q^{\pi^*}$.

Let

$$\mu(s,a) = \underset{\mathbb{P}(\tau)}{\mathbb{E}} \left[\sum_{i=0}^{k-1} \alpha_i \underset{s' \sim \tau_i(s,a), a' \sim \pi(\cdot|s')}{\mathbb{E}} [Q^{\pi^*}(s',a')]\right] = \underset{\tau \sim \widetilde{\mathbb{P}}(\tau)}{\mathbb{E}} \left[\underset{s' \sim \tau(s,a), a' \sim \pi(\cdot|s')}{\mathbb{E}} [Q(s',a')]\right],$$

$$\mu_k(s,a) = \sum_{i=0}^{k-1} \alpha_i \underset{s' \sim \tau_i(s,a), a' \sim \pi(\cdot|s')}{\mathbb{E}} [Q^{\pi^*}(s',a')].$$

Then we have:

$$\|\mathcal{B}^\pi Q^{\pi^*} - \hat{\mathcal{B}}^\pi Q^{\pi^*}\|_\infty = \gamma \sup_{s,a} |\mu_k(s,a) - \mu(s,a)|.$$

Noted that $|Q^{\pi^*}(s,a)| \leq \frac{R_{\max}}{1-\gamma}$. Let $C = \frac{2R_{\max}}{1-\gamma}$. According to the Hoeffding's Inequality, we have:

$$P(\sup_{s,a} |\mu_k(s,a) - \mu(s,a)| \geq \epsilon) = P(\exists (s,a), \quad s.t. |\mu_k(s,a) - \mu(s,a)| \geq \epsilon)$$

$$\leq \sum_{s,a} P(|\mu_k(s,a) - \mu(s,a)| \geq \epsilon)$$

$$\leq 2|S||A| \exp\left(-\frac{2\epsilon^2}{C^2 \sum_{i=0}^{k-1} \alpha_i^2}\right).$$

Let $\delta = 2|S||A| \exp\left(-\frac{2\epsilon^2}{C^2 \sum \alpha_i^2}\right)$. Solving for $\epsilon$ yields:

$$\epsilon = C\sqrt{\frac{\sum_{i=0}^{k-1} \alpha_i^2 \ln(2|S||A|/\delta)}{2}}.$$

Recalling that $\|\mathcal{B}^\pi Q^{\pi^*} - \hat{\mathcal{B}}^\pi Q^{\pi^*}\|_\infty = \gamma \sup_{s,a} |\mu_k(s,a) - \mu(s,a)|$, the operator error is bounded by $\gamma\epsilon$ with probability at least $1 - \delta$. Combining this with the contraction property:

$$\|Q^\pi - Q^{\pi^*}\|_\infty \leq \frac{1}{1-\gamma} \|\mathcal{B}^\pi Q^{\pi^*} - \hat{\mathcal{B}}^\pi Q^{\pi^*}\|_\infty \leq \frac{\gamma\epsilon}{1-\gamma}.$$

Substituting $\epsilon$ and $C = \frac{2R_{\max}}{1-\gamma}$ into the above, we conclude that with probability at least $1 - \delta$:

$$\|Q^\pi - Q^{\pi^*}\|_\infty \leq \frac{2\gamma R_{\max}}{(1-\gamma)^2} \sqrt{\frac{\sum_{i=0}^{k-1} \alpha_i^2}{2} \ln\left(\frac{2|S||A|}{\delta}\right)}.$$

Finally, since $|\mathbb{E}_{\rho_0,\pi}[Q^\pi] - \eta(\pi)| = |\mathbb{E}_{\rho_0,\pi}[Q^\pi] - \mathbb{E}_{\rho_0,\pi}[Q^{\pi^*}]| \leq \|Q^\pi - Q^{\pi^*}\|_\infty$, the following inequality holds with probability no less than $1 - \delta$:

$$|\mathbb{E}_{\rho_0,\pi}[Q^\pi] - \eta(\pi)| \leq \frac{2\gamma R_{\max}}{(1-\gamma)^2} \sqrt{\frac{\sum_{i=0}^{k-1} \alpha_i^2}{2} \ln\left(\frac{2|S||A|}{\delta}\right)}.$$

$\square$

**Lemma 1** (Lower-Bound). *If the true dynamics model $T$ satisfies $T \in \mathcal{T}$ but $T \notin \widetilde{\mathcal{T}}$, then $\eta(\pi)$ lower bounds the true performance $\eta(\pi, T)$, i.e., $\eta(\pi) \leq \eta(\pi, T)$.*

*Proof of Theorem 1.* Let $V^\pi(s)$ and $\tilde{V}^\pi(s)$ denote the value functions under the true dynamics $T$ and the pessimistic dynamics $\tilde{\tau}_i$, respectively. Their difference can be written as:

$$V^\pi(s) - \tilde{V}^\pi(s) = \mathop{\mathbb{E}}_{a\sim\pi(s)}\left[r(s,a) + \gamma\mathbb{E}_{s'\sim T}[V^\pi(s')] - r(s,a) - \gamma\mathbb{E}_{s'\sim\tilde{\tau}_i}[\tilde{V}^\pi(s')]\right]$$

Let:

$$\Delta(s_{t+1}) = \mathop{\mathbb{E}}_{s_{t+1}\sim T}[\tilde{V}^\pi(s_{t+1})] - \mathop{\mathbb{E}}_{s_{t+1}\sim\tilde{\tau}_i}[\tilde{V}^\pi(s_{t+1})].$$

Adding and subtracting $\gamma\mathbb{E}_{a\sim\pi(\cdot|s)}\left[\mathbb{E}_{s'\sim T}[\tilde{V}^\pi(s')]\right]$ on the right side of the equation, and rearranging, we obtain a recursive expression for the difference:

$$V^\pi(s) - \tilde{V}^\pi(s) = \mathop{\mathbb{E}}_{a\sim\pi(\cdot|s)}\left[\gamma\mathop{\mathbb{E}}_{s'\sim T}[\tilde{V}^\pi(s')] - \gamma\mathop{\mathbb{E}}_{s'\sim\tilde{\tau}_i}[\tilde{V}^\pi(s')]\right]$$
$$+ \gamma\mathop{\mathbb{E}}_{a\sim\pi(\cdot|s),s'\sim T}\left[V^\pi(s') - \tilde{V}^\pi(s')\right].$$

Then we have:

$$\eta(\pi, T) - \eta(\pi) = \mathop{\mathbb{E}}_{\pi,T}[\sum_{t=0}^\infty \gamma^{t+1}\Delta(s_{t+1})].$$

Recall the definition of the $q_{\tau_i}$:

$$q_{\tau_i}(s, a) = \mathop{\mathbb{E}}_{s'\sim\tau_i,a'\sim\pi(\cdot|s')}[Q(s',a')] = \mathop{\mathbb{E}}_{s'\sim\tau_i}[V(s')].$$

According to the definition of bottom-$k$ and $T \notin \widetilde{\mathcal{T}}$, we have $q_T \geq \max_{\tilde{\tau}_i\in\widetilde{\mathcal{T}}}\{q_{\tilde{\tau}_i}\}$ $\quad \forall s \in \mathcal{S}, a \in \mathcal{A}$. Then:

$$\Delta \geq q_T - \max_{\tilde{\tau}_i\in\widetilde{\mathcal{T}}}\{q_{\tilde{\tau}_i}\} \geq 0.$$

Finally, we get $\eta(\pi) \leq \eta(\pi, T)$. $\square$

**Lemma 2** (Dynamics Model Error). *The inequality for the difference between two Q-functions under different dynamics can be expressed as:*

$$\left|Q_{T_1}^\pi(s,a) - Q_{T_2}^\pi(s,a)\right| \leq \frac{2\gamma R_{\max}}{(1-\gamma)^2}d_{TV}(T_1, T_2),$$

*where $d_{TV}(T_1, T_2) = \max_{s,a} d_{TV}(T_1(\cdot|s,a), T_2(\cdot|s,a))$.*

*Proof of Lemma 2.* From the Bellman equation, it follows that:

$$Q_{T_1}^{\pi}(s,a) = \mathop{\mathbb{E}}_{s' \sim T_1(s,a)} \left[ r(s,a) + \gamma[V_{T_1}^{\pi}(s')] \right],$$

$$Q_{T_2}^{\pi}(s,a) = \mathop{\mathbb{E}}_{s' \sim T_2(s,a)} \left[ r(s,a) + \gamma[V_{T_2}^{\pi}(s')] \right].$$

Subtracting the two equations gives:

$$Q_{T_1}^{\pi} - Q_{T_2}^{\pi} = \gamma(\mathbb{E}_{T_1}[V_{T_1}^{\pi}] - \mathbb{E}_{T_1}[V_{T_2}^{\pi}]) + \gamma(\mathbb{E}_{T_1}[V_{T_2}^{\pi}] - \mathbb{E}_{T_2}[V_{T_2}^{\pi}]).$$

Then we have:

$$|Q_{T_1}^{\pi} - Q_{T_2}^{\pi}| \leq \gamma|\mathbb{E}_{T_1}[V_{T_1}^{\pi} - V_{T_2}^{\pi}]| + \gamma|\mathbb{E}_{T_1}[V_{T_2}^{\pi}] - \mathbb{E}_{T_2}[V_{T_2}^{\pi}]|.$$

According to the property of total variation distance, for any bounded function $f$ and two probability distributions $P, \widehat{P}$, we have:

$$\left| \mathop{\mathbb{E}}_{x \sim P}[f(x)] - \mathop{\mathbb{E}}_{x \sim \widehat{P}}[f(x)] \right| \leq (\sup f - \inf f) \cdot d_{TV}(P, \widehat{P}).$$

In our setting, the function $f$ is the value function $V_{T_2}^{\pi}$. Given that the reward satisfies $|r| \leq R_{\max}$, the value function is also bounded:

$$V_{\max} = \frac{R_{\max}}{1 - \gamma}, \quad V_{\min} = -\frac{R_{\max}}{1 - \gamma}$$

Thus, we have:

$$\gamma\left|\mathbb{E}_{T_1}[V_{T_2}^{\pi}] - \mathbb{E}_{T_2}[V_{T_2}^{\pi}]\right| \leq \frac{2\gamma R_{\max}}{1 - \gamma} d_{TV}(T_1(\cdot|s,a), T_2(\cdot|s,a)).$$

This implies that:

$$|Q_{T_1}^{\pi}(s,a) - Q_{T_2}^{\pi}(s,a)| \leq \gamma\|V_{T_1}^{\pi} - V_{T_2}^{\pi}\|_{\infty} + \frac{2\gamma R_{\max}}{1 - \gamma} d_{TV}(T_1, T_2).$$

Since the inequality holds for all $(s,a)$, it follows that:

$$\|Q_{T_1}^{\pi} - Q_{T_2}^{\pi}\|_{\infty} \leq \gamma\|V_{T_1}^{\pi} - V_{T_2}^{\pi}\|_{\infty} + \frac{2\gamma R_{\max}}{1 - \gamma} \max_{s,a} d_{TV}(T_1(\cdot|s,a), T_2(\cdot|s,a)).$$

Recall that $\|V^{\pi}\|_{\infty} \leq \|Q^{\pi}\|_{\infty}$, we obtain:

$$\|Q_{T_1}^{\pi} - Q_{T_2}^{\pi}\|_{\infty} \leq \gamma\|Q_{T_1}^{\pi} - Q_{T_2}^{\pi}\|_{\infty} + \frac{2\gamma R_{\max}}{1 - \gamma} d_{TV}(T_1, T_2).$$

After rearrangement and simplification, the following bound on the Q-function difference is obtained:

$$(1 - \gamma)\|Q_{T_1}^{\pi} - Q_{T_2}^{\pi}\|_{\infty} \leq \frac{2\gamma R_{\max}}{1 - \gamma} d_{TV}(T_1, T_2),$$

$$\|Q_{T_1}^{\pi} - Q_{T_2}^{\pi}\|_{\infty} \leq \frac{2\gamma R_{\max}}{(1 - \gamma)^2} d_{TV}(T_1, T_2).$$

This implies that:

$$|Q_{T_1}^{\pi}(s,a) - Q_{T_2}^{\pi}(s,a)| \leq \frac{2\gamma R_{\max}}{(1 - \gamma)^2} d_{TV}(T_1, T_2).$$

$\square$

**Theorem 2** (Approximate Lower-Bound). *Let the event $\mathcal{E} \triangleq \{\tau \in \mathcal{T}\}$ occur with probability at least 1-$\delta$. Then, the expected gap between $\eta(\pi)$ and $\eta(\pi, \tau)$ satisfies:*

$$\mathbb{E}_{\mathcal{T}}\big[\eta(\pi) - \eta(\pi, \tau)\big] \leq \frac{2\delta\gamma^2 R_{\max}}{(1-\gamma)^3}d_{\max} + \frac{\lambda\gamma\log k}{1-\gamma},$$

*where $d_{\max} = \sup\limits_{s.t.\neg\mathcal{E}} d_{\mathrm{TV}}(\tau, \tau_{proj})$ denotes the supremum of the total variation distance between $\tau$ and its projection onto the pessimistic subset $\widetilde{\mathcal{T}}$.*

*Proof of Theorem 2.* According to total expectation, we obtain:

$$\mathbb{E}_{\mathcal{T}}[\eta(\pi, \tau) - \eta(\pi)] = P(\mathcal{E})\mathbb{E}_{\mathcal{T}}[\eta(\pi, \tau) - \eta(\pi)|\mathcal{E}]$$
$$+ P(\neg\mathcal{E})\mathbb{E}_{\mathcal{T}}[\eta(\pi, \tau) - \eta(\pi)|\neg\mathcal{E}].$$

According to Lemma 1, we have:

$$\eta(\pi, \tau) - \eta(\pi) = \mathbb{E}_{\pi,\tau}\big[\sum_{t=0}^{\infty} \gamma^{t+1}\Delta(s_{t+1})\big],$$

where

$$\Delta(s_t) = \mathbb{E}_{s'\sim\tau}[\tilde{V}^\pi(s')] - \mathbb{E}_{s'\sim\tilde{\tau}_i}[\tilde{V}^\pi(s')]$$
$$= q_\tau(s'_t, a'_t) - \sum_{\tau_i\in\widetilde{\mathcal{T}}} \alpha_i q_{\tau_i}(s'_t, a'_t).$$

According to optimization problem:

$$\min_{\alpha_i} \sum_{i=0}^{k-1} \alpha_i q_{\tau_i}(s, a) - \lambda\mathcal{H}(\tilde{\alpha}),$$

where $\mathcal{H}(\tilde{\alpha})$ denotes entropy and $\sum_{i=0}^{k-1} \alpha_i = 1$.

We have:

$$q_\tau(s_t, a_t) - \sum_{i=0}^{k-1} \alpha_i q_{\tau_i}(s_t, a_t) \geq \lambda(\mathcal{H}(\omega) - \mathcal{H}(\tilde{\alpha})),$$

where $\omega$ is a one-hot vector and $\mathcal{H}(\omega) = 0$. Noticed that $\mathcal{H}(\tilde{\alpha}) \leq \log k$, it follows that:

$$\Delta(s_t) \geq -\lambda(\mathcal{H}(\tilde{\alpha})) \geq -\lambda\log k.$$

Then we have:

$$\mathbb{E}_{\mathcal{T}}[\eta(\pi, \tau) - \eta(\pi)|\mathcal{E}] \geq -\frac{\lambda\gamma\log k}{1-\gamma}.$$

Now, we address the second term. Let:

$$\Delta(s_t) = \mathbb{E}_{a_t\sim\pi(\cdot|s_t)}\Bigg[\big(q_\tau - q_{\tau_{\mathrm{proj}}}\big) + \big(q_{\tau_{\mathrm{proj}}} - \sum_{i=0}^{k-1} \alpha_i q_{\tau_i}(s_t, a_t)\big)\Bigg],$$

where $\tau_{\mathrm{proj}}$ is the projection of $T$ onto the subspace spanned by the pessimistic subset. Note that since the weighted coefficients $\alpha$ sum to one, the resulting subspace is in fact the convex hull, denoted by $\mathrm{conv}(\widetilde{\mathcal{T}})$. Therefore, $q_{\tau_{\mathrm{proj}}}$ can be expressed as:

$$q_{\tau_{\mathrm{proj}}} = \sum_{i=0}^{k-1} \omega_i q_{\tau_i},$$

where $\omega_i \geq 0$ and $\sum_{i=0}^{k-1} \omega_i = 1$. Then:

$$q_{\tau_{\text{proj}}} - \sum_{i=0}^{k-1} \alpha_i q_{\tau_i}(s_t, a_t) = \sum_{i=0}^{k-1} \omega_i q_{\tau_i} - \sum_{i=0}^{k-1} \alpha_i q_{\tau_i}(s_t, a_t).$$

According to aforementioned analysis, we have:

$$\sum_{i=0}^{k-1} \omega_i q_{\tau_i} - \sum_{i=0}^{k-1} \alpha_i q_{\tau_i}(s_t, a_t) \geq \lambda(\mathcal{H}(\omega) - \mathcal{H}(\tilde{\alpha})).$$

We know that $\mathcal{H}(\omega) \geq 0$ and $\mathcal{H}(\tilde{\alpha}) \leq \log k$. Therefore:

$$\sum_{i=0}^{k-1} \omega_i q_{\tau_i} - \sum_{i=0}^{k-1} \alpha_i q_{\tau_i}(s_t, a_t) \geq -\lambda \log k.$$

As for $q_T - q_{\tau_{\text{proj}}}$. According to Lemma 2, we obtain:

$$|q_\tau - q_{\tau_{\text{proj}}}| \leq \frac{2\gamma R_{\max}}{(1-\gamma)^2} d_{\max},$$

where $d_{\max} = \sup\limits_{s.t. \neg \mathcal{E}} d_{\text{TV}}(\tau, \tau_{proj})$ denotes the supremum of the total variation distance between $\tau$ and its projection onto the pessimistic subset $\widetilde{\mathcal{T}}$.

Finally, we get:

$$\mathbb{E}_{\mathcal{T}}\big[\eta(\pi, \tau) - \eta(\pi)\big] \geq -\frac{2\delta\gamma^2 R_{\max}}{(1-\gamma)^3} d_{\max} - \frac{\lambda\gamma \log k}{1-\gamma}.$$

We can rewrite this as:

$$\mathbb{E}_{\mathcal{T}}\big[\eta(\pi) - \eta(\pi, \tau)\big] \leq \frac{2\delta\gamma^2 R_{\max}}{(1-\gamma)^3} d_{\max} + \frac{\lambda\gamma \log k}{1-\gamma}.$$

$\square$

**Lemma 3** (Isotonicity). *Let $\Delta_{\max}$ be the maximum possible span of the Q-values across the model ensemble, i.e., $\Delta_{\max} \geq \max_{i,j} |q_{\tau_i}(s, a) - q_{\tau_j}(s, a)|$. If $\lambda \geq \Delta_{\max}$, then the operator $\mathcal{B}^\pi$ is isotonic. That is, for any $Q_1, Q_2 \in \mathbb{R}^{|S| \times |A|}$ such that $Q_1(s, a) \leq Q_2(s, a)$ for all $(s, a)$, we have:*

$$(\mathcal{B}^\pi Q_1)(s, a) \leq (\mathcal{B}^\pi Q_2)(s, a), \quad \forall(s, a).$$

*Proof.* The operator is defined as $(\mathcal{B}^\pi Q)(s, a) = r(s, a) + \gamma f(\mathbf{q}(s, a; Q))$, where $f(\mathbf{q}) = \sum_{i=1}^{k} \alpha_i(\mathbf{q}) q_i$ involves the Boltzmann weights $\alpha_i(\mathbf{q}) \propto \exp(-q_i/\lambda)$.

Since the mapping from $Q$ to the vector $\mathbf{q}$ is monotonic, it suffices to show that the aggregation function $f(\mathbf{q})$ is monotonic with respect to each component $q_m$ of the vector $\mathbf{q}$. This is equivalent to showing that the partial derivatives are non-negative:

$$\frac{\partial f(\mathbf{q})}{\partial q_m} \geq 0, \quad \forall m \in \{1, \ldots, k\}.$$

First, we derive the partial derivative of $f(\mathbf{q})$ with respect to an arbitrary component $q_m$. Using the derivative of the softmax

function $\frac{\partial \alpha_i}{\partial q_m} = -\frac{1}{\lambda}\alpha_i(\delta_{im} - \alpha_m)$, we have:

$$
\begin{aligned}
\frac{\partial f}{\partial q_m} &= \frac{\partial}{\partial q_m}\left(\sum_{i=0}^{k-1}\alpha_i q_i\right) \\
&= \alpha_m + \sum_{i=0}^{k-1} q_i \frac{\partial \alpha_i}{\partial q_m} \\
&= \alpha_m - \frac{1}{\lambda}\sum_{i=0}^{k-1} q_i \alpha_i (\delta_{im} - \alpha_m) \\
&= \alpha_m - \frac{1}{\lambda}\left(q_m \alpha_m - \alpha_m \sum_{i=0}^{k-1}\alpha_i q_i\right) \\
&= \alpha_m\left(1 - \frac{q_m - f(\mathbf{q})}{\lambda}\right).
\end{aligned}
$$

Since $\alpha_m > 0$ always holds, the sign of the derivative depends entirely on the term in the parentheses. To ensure $\frac{\partial f}{\partial q_m} \geq 0$, we require:

$$
1 - \frac{q_m - f(\mathbf{q})}{\lambda} \geq 0 \implies \lambda \geq q_m - f(\mathbf{q}).
$$

We observe that $f(\mathbf{q})$ is a convex combination of the elements in $\mathbf{q}$. Then, $f(\mathbf{q}) \geq \min_i q_i$. Therefore, the deviation of any single element from the mean is bounded by the span of the vector:

$$
q_m - f(\mathbf{q}) \leq q_m - \min_i q_i \leq \max_i q_i - \min_i q_i = \Delta(\mathbf{q}).
$$

By the assumption of the lemma, we have $\lambda \geq \Delta_{\max} \geq \Delta(\mathbf{q})$. Combining this with the inequality above:

$$
\lambda \geq \Delta(\mathbf{q}) \geq q_m - f(\mathbf{q}).
$$

Consequently, $1 - \frac{q_m - f(\mathbf{q})}{\lambda} \geq 0$, which implies $\frac{\partial f}{\partial q_m} \geq 0$ for all $m$.

Since all partial derivatives are non-negative, the function $f(\mathbf{q})$ is monotonic non-decreasing with respect to $\mathbf{q}$. Combined with the monotonicity of the expectation operator, we conclude that $Q_1 \leq Q_2 \implies \mathcal{B}^\pi Q_1 \leq \mathcal{B}^\pi Q_2$. $\qquad\square$

**Theorem 3** (Monotonicity of Pessimism). *Let $Q^\pi$ denote the fixed point of the Hybrid Belief Bellman Evaluation Operator. Then:*

- *For fixed $N$, the $Q^\pi$ is monotonically non-decreasing as the size of the pessimistic subset $k$ increases.*

- *For fixed $k$, the $\mathbb{E}_{\mathcal{T}}[Q^\pi]$ is monotonically non-increasing as the ensemble size $N$ increases.*

*Proof of Theorem 3.* Let $Q_{N,k}(s,a)$ and $\mathcal{B}^\pi_{N,k}$ denote the Q-function and Hybrid Belief Bellman Evaluation Operator induced by a model ensemble $\mathcal{T}$ of size $N$ and a pessimistic subset $\widetilde{\mathcal{T}}$ with size $k$, respectively. We now apply Lemma 3 to prove this theorem.

$\boxed{\text{Case1 : fixed N}}$

We first prove that when the pessimistic subset changes from bottom-$k$ ($\widetilde{\mathcal{T}}_k$) to bottom-$(k+1)$ ($\widetilde{\mathcal{T}}_{k+1}$), the corresponding Q-value is non-decreasing.

For simplicity, we temporarily omit the Q-function $(s,a)$, and use $q_i$ to represent the Q-value of the $i$-th most pessimistic model, satisfying $q_1 \leq q_2 \leq \cdots \leq q_N$.

- For the bottom-$k$ case, its Q-value is $S_k$:

$$S_k = \sum_{i=1}^{k} \alpha_{k,i} q_i = \frac{\sum_{j=1}^{k} q_j \exp(-q_j/\lambda)}{\sum_{i=1}^{k} \exp(-q_i/\lambda)}.$$

- For the bottom-$(k+1)$ case, its Q-value is $S_{k+1}$:

$$S_{k+1} = \sum_{i=1}^{k+1} \alpha_{k+1,i} q_i = \frac{\sum_{j=1}^{k+1} q_j \exp(-q_j/\lambda)}{\sum_{i=1}^{k+1} \exp(-q_i/\lambda)}.$$

Let $N_k = \sum_{i=j}^{k} q_j \exp(-q_j/\lambda)$ be the numerator of $S_k$, and $Z_k = \sum_{i=1}^{k} \exp(-q_i/\lambda)$ be the denominator (normalization factor) of $S_k$. Then $S_k = N_k/Z_k$.

The expression for $S_{k+1}$ can be formulated using $N_k$, $Z_k$, and the newly introduced term:

$$S_{k+1} = \frac{N_k + q_{k+1} \exp(-q_{k+1}/\lambda)}{Z_k + \exp(-q_{k+1}/\lambda)}.$$

Noted that $q_{k+1} \geq S_k$, i.e., $q_{k+1} \geq \frac{N_k}{Z_k}$. We obtain:

$$Z_k q_{k+1} \exp(-q_{k+1}/\lambda) \geq N_k \exp(-q_{k+1}/\lambda).$$

Adding $Z_k N_k$ to both sides of the equation and observing that $Z_k > 0$, we obtain:

$$\frac{N_k + q_{k+1} \exp(-q_{k+1}/\lambda)}{Z_k + \exp(-q_{k+1}/\lambda)} \geq \frac{N_k}{Z_k}.$$

It equals that:

$$S_{k+1} \geq S_k.$$

By recursively applying Lemma 3, we finally obtain:

$$Q_{N,k+1}^{\pi}(s,a) \geq Q_{N,k}^{\pi}(s,a).$$

---

Case2 : fixed k

Let $\mathcal{T}_N$ denote a specific ensemble of $N$ dynamics models sampled i.i.d. from the prior belief $\mathbb{P}(\tau)$. Let $f(\mathcal{T}_N) \triangleq Q_{|\mathcal{T}|,k}^{\pi}$ be the Q-function fixed point computed via the Hybrid Belief Bellman Evaluation Operator. Since $\mathcal{T}_N$ is a random variable, $f(\mathcal{T}_N)$ is also a random variable. We aim to show that:

$$\mathbb{E}_{\mathcal{T}_{N+1}}[f(\mathcal{T}_{N+1})] \leq \mathbb{E}_{\mathcal{T}_N}[f(\mathcal{T}_N)].$$

To prove this, it is sufficient to consider a fixed ensemble $\mathcal{T}_N$ of size $N$ and a new, randomly sampled model $\tau' \sim \mathbb{P}(\tau)$. We can then form a new ensemble of size $N+1$ as $\mathcal{T}_{N+1} = \mathcal{T}_N \cup \{\tau'\}$. If we can show that the following inequality holds for any fixed $\mathcal{T}_N$:

$$\mathbb{E}_{\tau' \sim \mathbb{P}(\tau)}[f(\mathcal{T}_N \cup \{\tau'\})] \leq f(\mathcal{T}_N),$$

then taking the expectation over $\mathcal{T}_N$ on both sides yields the desired result.

Let $\widetilde{\mathcal{T}}_k(\mathcal{S})$ denote the pessimistic subset of size $k$ selected from a larger set $\mathcal{S}$. Let $q_k(\mathcal{T}_N)$ be the $k$-th smallest Q-value among the models in $\mathcal{T}_N$. For different scenarios, we provide the following analysis.

**Case A**: The new model is optimistic ($q' > q_k(\mathcal{T}_N)$).

In this case, the new model $\tau'$ is not among the $k$ most pessimistic models in the combined set $\mathcal{T}_N \cup \{\tau'\}$. Therefore, the pessimistic subset $\widetilde{\mathcal{T}}$ selected from the new, larger ensemble is identical to the one selected from the old ensemble:

$$\widetilde{\mathcal{T}}_k(\mathcal{T}_N \cup \{\tau'\}) = \widetilde{\mathcal{T}}_k(\mathcal{T}_N).$$

Since the pessimistic subset $\widetilde{\mathcal{T}}$ over which the Bellman operator is defined remains unchanged, the resulting Q-function fixed point is also identical:

$$f(\mathcal{T}_N \cup \{\tau'\}) = f(\mathcal{T}_N).$$

**Case B**: The new model is pessimistic ($q' \leq q_k(\mathcal{T}_N)$).

In this case, the new model $\tau'$ is pessimistic enough to be included in the bottom-$k$ subset. It will replace the "least pessimistic" model (the one with value $q_k(\mathcal{T}_N)$) from the original subset $\widetilde{\mathcal{T}}_k(\mathcal{T}_N)$. The new pessimistic subset $\widetilde{\mathcal{T}}_k(\mathcal{T}_N \cup \{\tau'\})$ is therefore stochastically more pessimistic than the original subset $\widetilde{\mathcal{T}}_k(\mathcal{T}_N)$. Based on the conclusions of the aforementioned analysis, we have:

$$f(\mathcal{T}_N \cup \{\tau'\}) \leq f(\mathcal{T}_N).$$

Let $S(\mathbf{q}) = \sum_{i=0}^{k-1} \alpha_i(\mathbf{q}) q_i$ denote the weighted sum, where $\mathbf{q} \in \mathbb{R}^k$ represents the vector of Q-values in the pessimistic subset.

To prove $\mathbb{E}[f(\mathcal{T}_N \cup \{\tau'\})] \leq f(\mathcal{T}_N)$, it suffices to show that the operator $S(\mathbf{q})$ is monotonically non-decreasing with respect to each component $q_i$. If monotonicity holds, then $q' < q_{\max}$ implies $S(\ldots, q', \ldots) \leq S(\ldots, q_{\max}, \ldots)$, which proves the theorem. We compute the partial derivative of $S(\mathbf{q})$ with respect to an arbitrary component $q_j$. Recall that $\alpha_j = \frac{\exp(-q_j/\lambda)}{Z}$, where $Z = \sum_{l=0}^{k-1} \exp(-q_l/\lambda)$.

$$
\begin{aligned}
\frac{\partial S}{\partial q_j} &= \frac{\partial}{\partial q_j} \left( \frac{\sum_i q_i \exp(-q_i/\lambda)}{Z} \right) \\
&= \frac{\left( \exp(-q_j/\lambda) - \frac{q_j}{\lambda} \exp(-q_j/\lambda) \right) Z - \left( \sum_i q_i \exp(-q_i/\lambda) \right) \left( -\frac{1}{\lambda} \exp(-q_j/\lambda) \right)}{Z^2} \\
&= \frac{\exp(-q_j/\lambda)}{Z} \left[ 1 - \frac{q_j}{\lambda} + \frac{1}{\lambda} \underbrace{\frac{\sum_i q_i \exp(-q_i/\lambda)}{Z}}_{S(\mathbf{q})} \right] \\
&= \alpha_j \left( 1 - \frac{q_j - S(\mathbf{q})}{\lambda} \right).
\end{aligned}
$$

To ensure monotonicity ($\frac{\partial S}{\partial q_j} \geq 0$), the term in the parentheses must be non-negative. Specifically, we require that $\lambda \geq q_j - S(\mathbf{q})$. This condition is directly derived from the assumptions provided in Lemma 3.

*Justification:* Since $\widetilde{\mathcal{T}}$ consists of the $k$ smallest values from a larger ensemble, the values $q_i$ are concentrated at the lower tail of the distribution. $S(\mathbf{q})$ is their weighted average. Thus, the deviation $q_i - S(\mathbf{q})$ is naturally small. For a reasonable choice of $\lambda$ (not vanishingly small), this condition holds.

Since $q' < q_k(\mathcal{T}_N)$ (the new model is strictly more pessimistic than the one it replaces), and $S$ is monotonically non-decreasing, we conclude:

$$f(\mathcal{T}_N \cup \{\tau'\}) \leq f(\mathcal{T}_N).$$

Taking expectations yields the final result.

In both Case A and Case B, we show that for any possible realization of the new model $\tau'$, the following inequality holds:

$$f(\mathcal{T}_N \cup \{\tau'\}) \leq f(\mathcal{T}_N).$$

Since this holds for all outcomes of the random variable $\tau'$, it must also hold in expectation:

$$\mathbb{E}_{\tau'}[f(\mathcal{T}_N \cup \{\tau'\})] \leq \mathbb{E}_{\tau'}[f(\mathcal{T}_N)] = f(\mathcal{T}_N).$$

This inequality holds for any fixed ensemble $\mathcal{T}_N$. We now take the expectation over the random sampling of $\mathcal{T}_N$ on both sides:

$$\mathbb{E}_{\mathcal{T}_N}\left[ \mathbb{E}_{\tau'}[f(\mathcal{T}_N \cup \{\tau'\})] \right] \leq \mathbb{E}_{\mathcal{T}_N}[f(\mathcal{T}_N)].$$

By definition of i.i.d. sampling, the left-hand side is equivalent to the expectation over a model ensemble of size $N + 1$. Thus, we obtain the final conclusion:

$$\mathbb{E}_{\mathcal{T}_{N+1}}[Q_{N+1,k}^\pi] \le \mathbb{E}_{\mathcal{T}_N}[Q_{N,k}^\pi].$$

$\square$

**Theorem 4.** *The Optimal Hybrid Belief Bellman Operator $\widehat{\mathcal{B}}^*$ (Eq. (9)) is a contraction mapping. Repeatedly applying the operator $\widehat{\mathcal{B}}^*$ to any initial function $Q : \mathcal{S} \times \mathcal{A} \to \mathbb{R}$ yields a sequence converging to $Q^*$. The corresponding optimal policy is*

$$\pi^* = (\nabla\psi)^{-1}\big(\frac{1}{\beta}(Q^* - z) + \nabla\psi(\mu)\big),$$

*where $z$ denotes the Lagrange multiplier introduced to enforce probability normalization.*

*Proof of Theorem 4.* We now prove that $\widehat{\mathcal{B}}^*$ is a $\gamma$-contraction. For any $Q_1, Q_2 \in \mathbb{R}^{|\mathcal{S}| \times |\mathcal{A}|}$, let

$$V_1^*(s) = \max_\pi \big\{ \mathbb{E}_{a\sim\pi}[Q_1(s,a)] - \beta D_\psi(\pi(\cdot|s), \mu(\cdot|s)) \big\}$$

$$V_2^*(s) = \max_\pi \big\{ \mathbb{E}_{a\sim\pi}[Q_2(s,a)] - \beta D_\psi(\pi(\cdot|s), \mu(\cdot|s)) \big\}$$

According to definition, we have:

$$V_1^*(s) = \mathbb{E}_{a\sim\pi_1^*}[Q_1(s,a)] - \beta D_\psi(\pi_1^*, \mu),$$

$$V_2^*(s) \ge \mathbb{E}_{a\sim\pi_1^*}[Q_2(s,a)] - \beta D_\psi(\pi_1^*, \mu).$$

Then

$$\begin{aligned}
V_1^*(s) - V_2^*(s) &\le \big(\mathbb{E}_{\pi_1^*}[Q_1] - \beta D_\psi\big) - \big(\mathbb{E}_{\pi_1^*}[Q_2] - \beta D_\psi\big) \\
&= \mathbb{E}_{\pi_1^*}[Q_1(s,a) - Q_2(s,a)] \\
&\le \mathbb{E}_{\pi_1^*}[\|Q_1 - Q_2\|_\infty] \\
&= \|Q_1 - Q_2\|_\infty.
\end{aligned}$$

We have:

$$\|V_1^* - V_2^*\|_\infty \le \|Q_1 - Q_2\|_\infty.$$

Similar to the derivation of operator convergence in Theorem 1, we define:

$$x_i(Q) \triangleq \mathbb{E}_{s'\sim\tau_i}[V_Q^*(s')].$$

Then, we have: $(\widehat{\mathcal{B}}^*Q)(s,a) = r(s,a) + \gamma f(\mathbf{x}(Q))$, where $f(\mathbf{x}) = \sum_{i=0}^{k-1}\alpha_i(\mathbf{x})x_i$. For any $(s,a) \in |\mathcal{S}| \times |\mathcal{A}|$,

$$\begin{aligned}
|(\widehat{\mathcal{B}}^*Q_1)(s,a) - (\widehat{\mathcal{B}}^*Q_2)(s,a)| &= |(r(s,a) + \gamma f(\mathbf{x}(Q_1))) - (r(s,a) + \gamma f(\mathbf{x}(Q_2)))| \\
&= \gamma|f(\mathbf{x}(Q_1)) - f(\mathbf{x}(Q_2))| \\
&\le \gamma\left(1 + \frac{\Delta_{\max}}{2\lambda}\right)\|\mathbf{x}(Q_1) - \mathbf{x}(Q_2)\|_\infty.
\end{aligned}$$

We now consider the difference between the input vectors $\|\mathbf{x}(Q_1) - \mathbf{x}(Q_2)\|_\infty$. For any specific model $\tau_i$, the difference is expressed as:

$$\begin{aligned}
|x_i(Q_1) - x_i(Q_2)| &= \left| \mathbb{E}_{s'\sim\tau_i}[V_{Q_1}^*(s')] - \mathbb{E}_{s'\sim\tau_i}[V_{Q_2}^*(s')] \right| \\
&\le \mathbb{E}_{s'\sim\tau_i}\big[|V_{Q_1}^*(s') - V_{Q_2}^*(s')|\big] \\
&\le \max_{s'}|V_{Q_1}^*(s') - V_{Q_2}^*(s')| \\
&= \|V_{Q_1}^* - V_{Q_2}^*\|_\infty.
\end{aligned}$$

Then:

$$|(\widehat{\mathcal{B}}^* Q_1)(s,a) - (\widehat{\mathcal{B}}^* Q_2)(s,a)| \leq \gamma \left(1 + \frac{\Delta_{\max}}{2\lambda}\right) \|Q_1 - Q_2\|_\infty.$$

Since the above holds for all $(s,a)$, we take its supremum:

$$\|\widehat{\mathcal{B}}^* Q_1 - \widehat{\mathcal{B}}^* Q_2\|_\infty = \sup_{s,a} |(\widehat{\mathcal{B}}^* Q_1)(s,a) - (\widehat{\mathcal{B}}^* Q_2)(s,a)| \leq \gamma \left(1 + \frac{\Delta_{\max}}{2\lambda}\right) \|Q_1 - Q_2\|_\infty.$$

Based on the results in Section C.2, the optimal policy corresponding to the value function $V^*(s)$ is given by:

$$\pi^* = (\nabla\psi)^{-1}\left(\frac{1}{\beta}(Q^* - z) + \nabla\psi(\mu)\right).$$

We now show that $\pi^* = (\nabla\psi)^{-1}\left(\frac{1}{\beta}(Q^* - z) + \nabla\psi(\mu)\right)$ is the optimal policy. For any policy $\pi'$, according to definition of $V^*(s)$, we have:

$$V^*(s) \geq (\bar{\mathcal{B}}^{\pi'} V^*)(s),$$

where $\bar{\mathcal{B}}$ is Bellman evaluation operator. Then

$$V^* \geq \lim_{k\to\infty} (\bar{\mathcal{B}}^{\pi'})^k V^* = V^{\pi'}.$$

Since this inequality holds for any policy $\pi'$, we conclude that $\pi^* = (\nabla\psi)^{-1}\left(\frac{1}{\beta}(Q^* - z) + \nabla\psi(\mu)\right)$ is optimal for regularized objective function. $\square$

**Theorem 5** (Monotonic Improvement). *Starting from an arbitrary initial policy $\pi_0$, consider the sequence of policies $\{\pi_i\}$ generated by iteratively solving the Bregman-regularized subproblem: $\pi_{i+1} = (\nabla\psi)^{-1}\left(\frac{1}{\beta}(Q - z) + \nabla\psi(\pi_i)\right)$. Then it holds that: $\eta(\pi_{i+1}) \geq \eta(\pi_i)$.*

*Proof of Theorem 5.* Let $Q^\pi$ denote the fixed point of the policy $\pi$ under $\mathcal{B}^\pi$. The core idea is to show $Q^{\pi_{i+1}} \geq Q^{\pi_i}$. Firstly, we define regularized evaluation operator:

$$(\widehat{\mathcal{B}}^\pi Q)(s,a) = r(s,a) + \gamma \sum_{\tau_i \in \widetilde{\mathcal{T}}} \alpha_i \mathop{\mathbb{E}}_{s'\sim\tau_i, a'\sim\pi} \left[Q(s',a') - \beta D_\psi(\pi(\cdot|s'), \mu(\cdot|s'))\right].$$

Let $Q^\pi_\mu$ denote the fixed point of $\widehat{\mathcal{B}}^\pi$, i.e., $Q^\pi_\mu = \widehat{\mathcal{B}}^\pi Q^\pi_\mu$. Then, we have:

$$(\mathcal{B}^\pi Q^{\pi_{i+1}}_{\pi_i})(s,a) - (\widehat{\mathcal{B}}^\pi Q^{\pi_{i+1}}_{\pi_i})(s,a) = \gamma \mathop{\mathbb{E}}_{s',\pi} \left[\beta D_\psi\big(\pi_{i+1}(\cdot|s')\|\pi_i(\cdot|s')\big)\right].$$

Noticed that $D_\psi\big(\pi_{i+1}(\cdot|s')\|\pi_i(\cdot|s')\big) \geq 0$. According to Lemma 3, the converged Q-function also satisfies this inequality, i.e.,

$$Q^{\pi_{i+1}} \geq Q^{\pi_{i+1}}_{\pi_i}.$$

Next, we show that $Q^{\pi_{i+1}}_{\pi_i} \geq Q^{\pi_i}$. By definition, $\pi_{i+1}$ is the policy that maximizes the regularized objective function with respect to the reference policy $\pi_i$. Therefore, the value it achieves under this objective is greater than or equal to that of any other policy. Then we have:

$$\widehat{\eta}(\pi_{i+1};\pi_i) \geq \widehat{\eta}(\pi_i;\pi_i) = \eta(\pi_i).$$

It is equal to $Q^{\pi_{i+1}}_{\pi_i} \geq Q^{\pi_i}$.

Based on the analysis of both steps, we arrive at the following conclusion:

$$Q^{\pi_{i+1}} \geq Q^{\pi_i}.$$

$\square$

## B. Theoretic Results on Hybrid Belief

### B.1. The Optimal $\alpha$

We need to solve the following regularized optimization problem to find the optimal weight coefficients $\tilde{\alpha} = \{\alpha_0, \alpha_1, \ldots, \alpha_{k-1}\}$:

$$\min_{\tilde{\alpha}} \left\{ \sum_{i=0}^{k-1} \alpha_i q_{\tau_i}(s, a) - \lambda \mathcal{H}(\tilde{\alpha}) \right\},$$

$$\text{s.t.} \quad \sum_{i=0}^{k-1} \alpha_i = 1,$$

$$\alpha_i \geq 0, \quad \forall i \in \{0, 1, \ldots, k-1\},$$

where the Shannon entropy $\mathcal{H}(\tilde{\alpha}) = -\sum_{i=0}^{k-1} \alpha_i \log(\alpha_i)$. Substituting this into the objective function, the original problem is equivalent to:

$$\min_{\tilde{\alpha}} \left\{ \sum_{i=0}^{k-1} \alpha_i q_i + \lambda \alpha_i \log(\alpha_i) \right\},$$

which matches the optimization problem solved when defining the likelihood in the original paper.

For simplicity, we use $q_i$ to denote $q_{\tau_i}(s, a)$. The Lagrangian function $\mathcal{L}(\tilde{\alpha}, z, \tilde{\mu})$ is constructed as follows:

$$\mathcal{L}(\tilde{\alpha}, z, \tilde{\mu}) = \sum_{i=0}^{k-1} \alpha_i q_i + \lambda \sum_{i=0}^{k-1} \alpha_i \log(\alpha_i)$$

$$- z \left( \sum_{i=0}^{k-1} \alpha_i - 1 \right) - \sum_{i=0}^{k-1} \mu_i \alpha_i.$$

Here, $z$ is the Lagrange multiplier corresponding to the equality constraint $\sum \alpha_i = 1$, and $\mu_i$ are the Lagrange multipliers corresponding to the inequality constraints $\alpha_i \geq 0$.

Due to the presence of the entropy term $\log(\alpha_i)$, the optimal solution must lie in the interior of the probability simplex, i.e., $\alpha_i > 0$. According to the complementary slackness condition $\mu_i \alpha_i = 0$, when $\alpha_i > 0$, we must have $\mu_i = 0$. Under this condition, the stationarity condition simplifies to:

$$q_i + \lambda(\log(\alpha_i) + 1) - z = 0.$$

The value of $\alpha_i$ is determined by solving the equation:

$$\lambda \log(\alpha_i) = z - \lambda - q_i,$$

$$\log(\alpha_i) = \frac{z - \lambda}{\lambda} - \frac{q_i}{\lambda},$$

$$\alpha_i = \exp\left( \frac{z - \lambda}{\lambda} - \frac{q_i}{\lambda} \right)$$

$$= \exp\left( \frac{z - \lambda}{\lambda} \right) \cdot \exp\left( -\frac{q_i}{\lambda} \right).$$

Since $\exp\left( \frac{z-\lambda}{\lambda} \right)$ is a constant that does not depend on the index $i$, we can conclude that:

$$\alpha_i \propto \exp\left( -\frac{1}{\lambda} q_i \right).$$

Let $\alpha_i = C \cdot \exp\left(-\frac{1}{\lambda} q_i\right)$, then:

$$C = \frac{1}{\sum_{j=0}^{k-1} \exp\left(-\frac{1}{\lambda} q_j\right)}.$$

Substituting the constant $C$ back into the expression yields the final form of the optimal solution:

$$\alpha_i = \frac{\exp\left(-\frac{1}{\lambda} q_i\right)}{\sum_{j=0}^{k-1} \exp\left(-\frac{1}{\lambda} q_j\right)}.$$

By substituting the optimal solution $\alpha_i \propto \exp(-\frac{1}{\lambda} q_i)$ into the primal problem, the resulting optimal value is given by:

$$-\lambda \log\left(\sum_{j=0}^{k-1} \exp(-\frac{q_j}{\lambda})\right).$$

## C. Theoretic Results on Regularized Policy Iteration

### C.1. Property of Bregman Divergence

Mirror Descent refers to a class of optimization algorithms that generalize gradient descent by employing Bregman divergences to measure the proximity between successive iterates. Unlike standard gradient descent, which relies on Euclidean geometry, mirror descent adjusts the update direction by defining a notion of distance in the parameter space using Bregman divergence. Its update formula is:

$$\theta_{t+1} = \arg\max_{\theta \in \Theta} \left\{ \langle \nabla f(\theta_t), \theta \rangle - \beta D_\psi(\theta, \theta_t) \right\},$$

where:

- $f(\theta)$ is the objective function;

- $D_\psi(\theta, \theta_t)$ is the Bregman divergence based on the potential function $\psi(\theta)$, defined as:

$$D_\psi(\theta, \theta_t) = \psi(\theta) - \psi(\theta_t) - \langle \nabla \psi(\theta_t), \theta - \theta_t \rangle.$$

The following derivation shows that gradient descent with Bregman divergence regularization can be approximated as:
$\theta_{t+1} = \theta_t + \frac{1}{\beta} [\nabla^2 \psi(\theta_t)]^{-1} \nabla f(\theta_t)$.

**Local Approximation of Bregman Divergence.** The Bregman divergence $D_\psi(\theta, \theta_t)$ can be approximated by a Taylor expansion near $\theta_t$:

$$D_\psi(\theta, \theta_t) \approx \frac{1}{2}(\theta - \theta_t)^T \nabla^2 \psi(\theta_t)(\theta - \theta_t).$$

Thus, the local form of the Bregman divergence is equivalent to the quadratic norm defined by $\nabla^2 \psi(\theta_t)$.

**Update Formula.** Substitute the approximation of the Bregman divergence into the optimization problem:

$$\theta_{t+1} \approx \arg\max_\theta \left\{ \langle \nabla f(\theta_t), \theta \rangle - \frac{\beta}{2}(\theta - \theta_t)^T \nabla^2 \psi(\theta_t)(\theta - \theta_t) \right\}.$$

Taking the derivative of the objective function and setting it to zero, we get:

$$\nabla f(\theta_t) - \beta \nabla^2 \psi(\theta_t)(\theta_{t+1} - \theta_t) = 0.$$

Solving for $\theta_{t+1}$, we finally obtain:

$$\theta_{t+1} = \theta_t + \frac{1}{\beta} [\nabla^2 \psi(\theta_t)]^{-1} \nabla f(\theta_t),$$

## C.2. The optimal solution to the value function $V^*(s)$

In the policy iteration, we need to solve the following constrained optimization problem to update from the reference policy $\mu$ to the new policy $\pi$:

$$\max_{\pi} \left\{ \mathbb{E}_{a \sim \pi} [Q(s,a)] - \beta D_{\psi}(\pi, \mu) \right\},$$

$$\text{s.t.} \quad \sum_a \pi(a) = 1.$$

Here, $Q(s,a)$ is the action value, and $D_{\psi}(x,y) = \psi(x) - \psi(y) - \langle \nabla \psi(y), x - y \rangle$ is the Bregman divergence generated by the strictly convex potential function $\psi$.

We construct the Lagrangian function $L(\pi, z)$ for this optimization problem as follows:

$$L(\pi, z) = \sum_a \pi(a) Q(s,a) - \beta \left( \psi(\pi) - \psi(\mu) - \sum_a \frac{\partial \psi(\mu)}{\partial \mu(a)} (\pi(a) - \mu(a)) \right)$$

$$- z \left( \sum_a \pi(a) - 1 \right).$$

where $z$ is the Lagrange multiplier corresponding to the constraint $\sum_a \pi(a) = 1$. To derive the stationarity condition, we compute the partial derivative of $L$ with respect to $\pi(a)$ and set it to zero:

$$\frac{\partial L}{\partial \pi(a)} = Q(s,a) - \beta \left( \frac{\partial \psi(\pi)}{\partial \pi(a)} - \frac{\partial \psi(\mu)}{\partial \mu(a)} \right) - z = 0.$$

This yields the following condition:

$$\beta \frac{\partial \psi(\pi^*)}{\partial \pi(a)} = Q(s,a) + \beta \frac{\partial \psi(\mu)}{\partial \mu(a)} - z.$$

We finally obtain:

$$\pi^* = (\nabla \psi)^{-1} \left( \frac{1}{\beta} (Q - z) + \nabla \psi(\mu) \right).$$

Here, $Q$ is the vector of Q-values, and all operations are element-wise. The scalar $z$ is a normalization constant determined by the constraint $\sum_a \pi^*(a) = 1$.

### C.2.1. EXAMPLE

In this document, we derive the explicit functional form for $\pi^*$ when the Bregman divergence is the Kullback-Leibler (KL) divergence, which is generated by the negative entropy potential function.

$$\psi(\pi) = \sum_{a \in \mathcal{A}} \pi(a) \log \pi(a).$$

To compute its gradient, $\nabla \psi(\pi)$, we take the partial derivative with respect to each component $\pi(a)$. We use the product rule for the derivative of $x \log x$:

$$\frac{d}{dx}(x \log x) = (1) \cdot \log x + x \cdot \left( \frac{1}{x} \right) = \log x + 1.$$

Therefore, the $a$-th component of the gradient is:

$$\frac{\partial \psi(\pi)}{\partial \pi(a)} = \log \pi(a) + 1.$$

In vector form, the gradient is:

$$\nabla \psi(\pi) = \log \pi + \mathbf{1},$$

where the logarithm is applied element-wise and $\mathbf{1}$ is a vector of ones.

Next, we find the inverse of the gradient map. Let $Y = \nabla\psi(\pi)$. We want to solve for $\pi$ in terms of $Y$ to find the function $\pi = (\nabla\psi)^{-1}(Y)$.

$$Y = \log\pi + \mathbf{1},$$
$$Y - \mathbf{1} = \log\pi,$$
$$\exp(Y - \mathbf{1}) = \pi.$$

Thus, the inverse gradient map is:

$$(\nabla\psi)^{-1}(Y) = \exp(Y - \mathbf{1}).$$

Now we substitute these specific forms back into the general solution for $\pi^*$.

$$\pi^* = (\nabla\psi)^{-1}\left(\frac{1}{\beta}(Q - z) + \nabla\psi(\mu)\right)$$

$$= (\nabla\psi)^{-1}\left(\frac{1}{\beta}(Q - z) + (\log\mu + \mathbf{1})\right)$$

$$= \exp\left(\left[\frac{1}{\beta}(Q - z) + \log\mu + \mathbf{1}\right] - \mathbf{1}\right)$$

$$= \exp\left(\frac{1}{\beta}(Q - z) + \log\mu\right).$$

We can simplify this expression further using the properties of exponents. For each component $a$:

$$\pi^*(a) = \exp\left(\frac{Q(s, a) - z}{\beta} + \log\mu(a)\right)$$

$$= \exp\left(\frac{Q(s, a) - z}{\beta}\right) \cdot \exp(\log\mu(a))$$

$$= \mu(a) \cdot \exp\left(\frac{Q(s, a)}{\beta}\right) \cdot \exp\left(-\frac{z}{\beta}\right).$$

Here, the term $\exp(-z/\beta)$ is constant with respect to the action for a given state $s$. This ensures the policy is properly normalized. The resulting explicit form is:

$$\pi^*(a) = \frac{\mu(a)\exp\left(\frac{Q(s,a)}{\beta}\right)}{\sum_{a'\in\mathcal{A}}\mu(a')\exp\left(\frac{Q(s,a')}{\beta}\right)}.$$

## D. Experimental Details

### D.1. Dataset

#### D.1.1. D4RL

To comprehensively assess performance across diverse offline scenarios, our experimental assessment encompasses a comprehensive benchmark spanning eighteen distinct experimental settings. These domains arise from the combination of three continuous control tasks (hopper, walker2d, and halfcheetah) with six diverse offline datasets that vary in quality and collection strategy.

- random: dataset collected by a randomly initialized policy.

- expert: dataset collected by a fully-trained SAC agent.

- medium: dataset collected by a policy achieving approximately 33% of the expert's performance.

- medium-expert: dataset composed of an equal mixture (50–50 split) of medium and expert data.

*Table 7.* Reference performance across different environments.

| Environment | Random Policy | Expert Policy |
|---|---|---|
| Halfcheetah | -280.18 | 12135.0 |
| Hopper | -20.27 | 3234.3 |
| Walker2d | 1.63 | 4592.3 |
| Offline optimal liquidation | 0.0 | 135.0 |

- medium-replay: dataset composed of the replay buffer of a policy trained until it reaches the performance level of the medium agent.

- full-replay: dataset composed of the complete replay buffer of the SAC agent.

### D.1.2. OPTIMAL LIQUIDATION

This challenging task is an adaptation of the Optimal Liquidation Problem (Bao & Liu, 2019) for offline RL, following the setup introduced by (Rigter et al., 2023). We provide a brief overview of the task below.

The agent's objective is to convert an initial holding of 100 units of currency A into currency B by a final time $T$, under a stochastically evolving exchange rate. At each timestep, the agent decides the proportion of its remaining currency A to exchange.

At each decision point, the agent faces a classic dilemma: secure a known, immediate gain or wait for a potentially larger, uncertain future reward. Opting to wait is a high-stakes gamble. While it offers the potential for higher returns from a favorable exchange rate movement, it also exposes the agent to substantial losses if the rate declines and does not recover. On the other hand, safer approaches focus on reducing risk. These include converting small amounts over time to average out price changes, or converting the entire amount at once to prevent losses from a future drop in the exchange rate.

**Markov Decision Process (MDP) Formulation.** The state $s_t = (t, m_t, p_t)$ is 3-dimensional, consisting of the current timestep $t \in \{0, 1, \dots, T-1\}$, the remaining amount of currency A $m_t \in [0, 100]$, and the current exchange rate $p_t \in [0, \infty)$. The initial exchange rate is drawn from $p_0 \sim \mathcal{N}(1, 0.05^2)$. The continuous action space is defined as $\mathcal{A} = [-1, 1]$, where $a_t > 0$ specifies the conversion proportion of inventory $m_t$, and $a_t \leq 0$ denotes a holding action. The reward equals the amount of Currency B realized at each step. The state transition dynamics are primarily governed by the evolution of the exchange rate $p_t$, which is modeled as an Ornstein-Uhlenbeck (OU) process:

$$dp_t = \theta(\mu - p_t)dt + \sigma dW_t,$$

where $W_t$ is a standard Wiener process. The process parameters are configured as $\theta = 0.05$ (rate of mean reversion), $\mu = 1.5$ (long-term equilibrium price), and $\sigma = 0.2$ (volatility coefficient).

**Offline Dataset Collection.** The dataset is constructed using a random behavioral policy. At each step, the policy selects a non-conversion decision with probability 0.8, and samples a conversion action with probability 0.2 by drawing the proportion uniformly from the feasible range.

### D.1.3. STANDARD REFERENCE PERFORMANCE

The normalized score benchmarks an algorithm's performance on a standardized scale. On this scale, a random policy is set to 0 and an expert policy is set to 100. A score greater than 100 thus signifies that the policy learned from the offline dataset outperforms the online-trained expert policy. The score is computed as follows:

$$\text{score}_{\text{algo}} = 100 \times \frac{\text{performance}_{\text{algo}} - \text{performance}_{\text{expert}}}{\text{performance}_{\text{expert}} - \text{performance}_{\text{random}}}.$$

The reference performance is reported in Table 7.

*Table 8.* Hyperparameters

| Parameter | Value |
|---|:---:|
| Dynamics model learning rate | $10^{-4}$ |
| Policy learning rate | $3 \cdot 10^{-5}$ |
| Critic (Q-value) learning rate | $3 \cdot 10^{-4}$ |
| discounted factor ($\gamma$) | 0.99 |
| Size of the model ensemble ($N$) | 10 |
| Size of the pessimistic subset ($k$) | 5 |
| Entropy coefficient ($\lambda$) | 0.33 |
| Potential function coefficient ($\omega$) | 0.9 |
| Bregman divergence coefficient ($\beta$) | 0.1 |
| Layer size of policy | 256 |
| Layer size of dynamics model | 512 |
| Batch size for dynamics model learning | 512 |
| Batch size for policy learning | 256 |
| Dynamics model training epochs | 1000 |
| Policy training steps | $2 \cdot 10^6$ |
| Maximal horizon of PhyB | 1000 |

### D.2. Hyperparameters

Table 8 presents the detailed hyperparameter configurations. For the simpler hopper environment, dynamics models with layer and batch sizes of 256 units are sufficient. However, the increased complexity of walker2d and halfcheetah requires larger architectures with 512-unit layers to ensure convergence. The selection of specific parameters, such as $\omega$, is based on heuristics.

### D.3. Dynamics model

Following standard practice, we parameterize dynamics models as neural networks that output Gaussian distributions over next states and rewards. We independently train 100 such models via maximum likelihood estimation, then randomly sample $N$ models to form an ensemble during prediction.

### D.4. Compute Infrastructure

Our computational experiments are conducted on a server equipped with four NVIDIA GeForce RTX 3090 Ti GPUs, an Intel(R) Xeon(R) Platinum 8383C CPU @ 2.70GHz, and 256 GB of system memory.

### D.5. Computational Overhead

Like other model-based methods, PhyB's training time scales linearly with ensemble size $N$, while inference (policy sampling) is computationally inexpensive. To ensure a fair comparison, we benchmark PhyB against model-based (PMDB, ADM) and model-free (DMG, TD3+BC) baselines on the same GPU with the same batch size, while maintaining consistent dynamics model parameters for the model-based methods. The GPU runtime for each method is reported, with results summarized in Table 10.

### D.6. How sensitive is the method to inaccurate value functions early in training?

While the subset selection depends on critic estimates, empirical results in Figure 1 and Figure 3 demonstrate that PhyB remains remarkably robust to initial value inaccuracies, with evaluation performance showing a steady, monotonic increase toward convergence.

To mitigate potential instability or overestimation, we augment the training of the value function by blending real transitions from the dataset with rollouts from the learned dynamics models.

Furthermore, our approach demonstrates inherent stability by using a fixed set of hyperparameters across all D4RL tasks

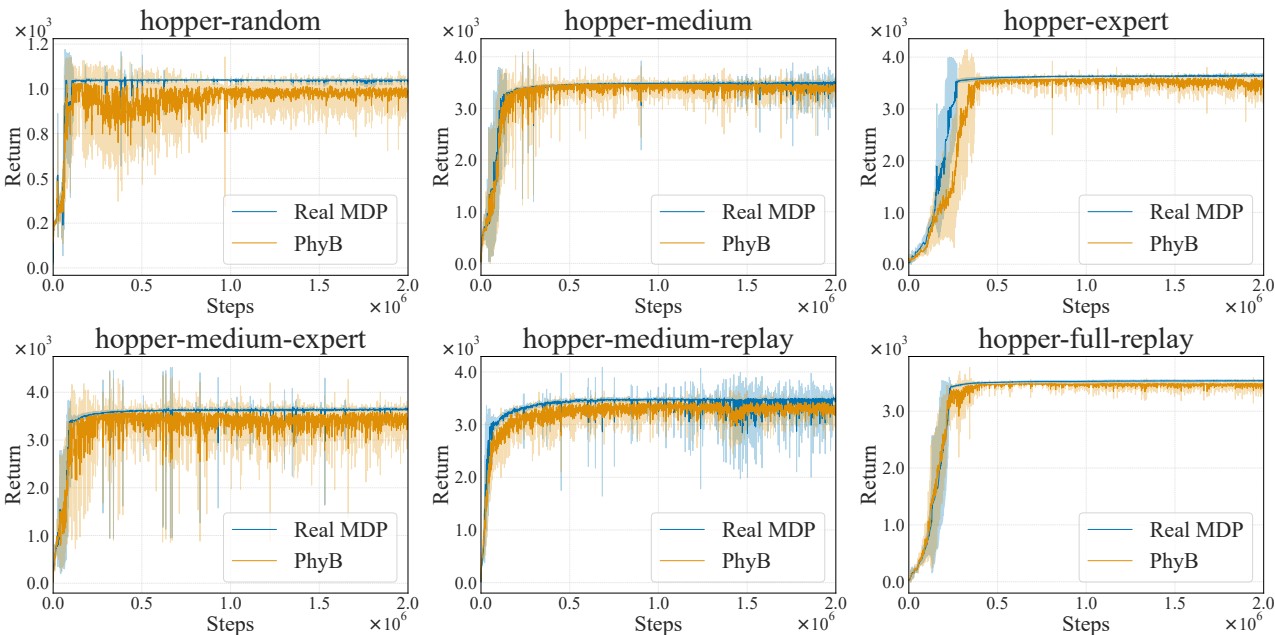

*Figure 3.* Learning and evaluation curves in Hopper-v2 environment.

*Table 9.* Additional D4RL results.

| Dataset-v2 | ADM | ORPO | 1R2R | MOBILE | PhyB |
|---|---|---|---|---|---|
| hopper-random | 32.7±0.2 | 9.2±1.4 | 30.9±2.3 | 31.9±0.6 | **33.9**±1.1 |
| halfcheetah-random | **45.4**±2.8 | 40.8±1.6 | 36.9±6.4 | 39.3±3.0 | 34.7±1.7 |
| walker2d-random | 22.2±0.2 | 10.8±9.3 | 7.6±12.2 | 17.9±6.6 | **23.5**±1.5 |
| hopper-medium | 107.4±0.6 | 30.4±37.4 | 80.2±15.8 | 106.6±0.6 | **109.4**±2.0 |
| halfcheetah-medium | 72.2±0.6 | 73.4±0.5 | 74.5±2.1 | **74.6**±1.2 | 74.5±1.8 |
| walker2d-medium | 93.2±1.1 | 55.5±23.4 | 63.9±31.8 | 87.7±1.1 | **95.5**±8.5 |
| hopper-medium-replay | 104.4±0.4 | 104.6±1.5 | 92.9±10.7 | 103.9±1.0 | **110.7**±1.3 |
| halfcheetah-medium-replay | 67.6±3.4 | 72.8±0.9 | 65.7±2.3 | 71.7±1.2 | **74.7**±1.5 |
| walker2d-medium-replay | **95.6**±2.1 | 91.1±2.0 | 92.2±2.3 | 89.9±1.5 | 85.4±3.9 |
| hopper-medium-expert | 112.7±0.3 | 111.0±0.6 | 81.6±22.9 | 112.6±0.2 | **116.5**±2.1 |
| halfcheetah-medium-expert | 103.7±0.2 | 101.5±3.1 | 96.0±6.0 | 108.2±2.5 | **109.4**±1.5 |
| walker2d-medium-expert | 114.9±0.3 | 108.8±3.2 | 90.9±5.9 | **115.2**±0.7 | 112.4±1.1 |
| Average Score | 81.0 | 67.5 | 67.8 | 80.0 | **81.7** |

without per-task tuning. This confirms the reliability of our ranking mechanism, even as the critic evolves.

### D.7. Addition Experimental Results

**Comparison with the recent SoTA algorithms.** Table 9 compares our method against recent state-of-the-art model-based approaches, including ADM (Lin et al., 2025), ORPO (Zhai et al., 2024), 1R2R (Rigter et al., 2023) and MOBILE (Sun et al., 2023). PhyB achieves superior performance on 8 out of 12 benchmarks and delivers competitive results on the remaining 4.

**Training curve.** Figure 3 compares the true return of the policy with the return estimated under PhyB in the Hopper environment, with evaluation performed every 1,000 steps. As shown in the figure, the return under PhyB closely tracks the trend of the true return and consistently remains below or equal to it, serving as a lower bound. The trend of the curves

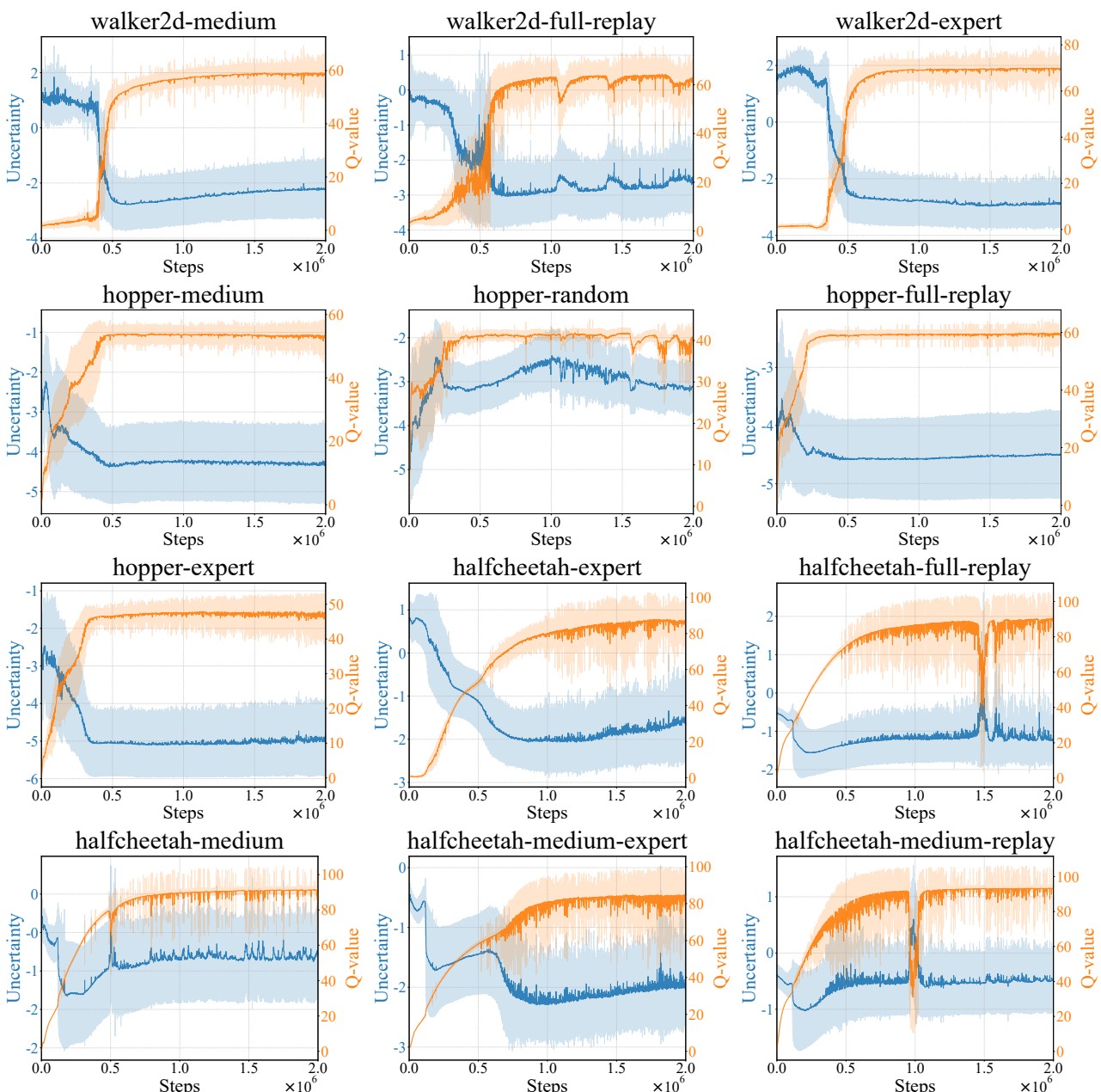

*Figure 4.* Evolution of Q-values and uncertainty for encountered state-action pairs during training.

experimentally validates Theorem 2). Moreover, we observe that performance improves nearly monotonically throughout training, providing empirical support for Theorem 5.

**Uncertainty quantification.** We quantify the uncertainty for a state-action pair $(s, a)$ by computing the log standard deviation of next-state predictions across the pessimistic subset $\widetilde{\mathcal{T}}$, as illustrated in Figure 4. The results in Figure 4 complement those reported in the original paper. We observe that a sharp increase in this uncertainty metric is consistently associated with a corresponding decrease in the Q-value, reinforcing the inverse relationship between predictive uncertainty and value estimation.

**Monotonicity.** As shown in Table 11, we conduct an ablation study on the impact of the pessimistic subset size on policy performance. The results serve as a supplement to the main text. Combining the findings from the main experiment with

*Table 10.* Computational overhead of various methods.

| Method | Train (ms/iteration) | Evaluation (ms/iteration) |
|---|---|---|
| PhyB ($N$=20) | 198.8428±0.9615 | 3.0978±0.0126 |
| PhyB ($N$=10) | 106.4067±0.3879 | 3.0744±0.0139 |
| PhyB ($N$=5) | 59.0712±0.3292 | 3.0889±0.0142 |
| PMDB | 82.3452±0.6088 | 2.9128±0.0886 |
| ADM | 29.7432±0.3419 | 9.0266±0.4126 |
| DMG | 11.3725±0.1962 | 1.6785±0.0051 |
| TD3+BC | 7.2683±0.1789 | 1.1516±0.0017 |

*Table 11.* Impact of the pessimistic subset $k$ on policy performance.

| $(N, k)$ | Walker2d-E | HalfCheetah-E |
|---|---|---|
| (10,3) | 110.4±1.7 | 106.1±1.2 |
| (10,5) | 116.3±1.1 | 113.7±1.0 |

*Table 12.* Ablation study on the effect of the weight averaging. "HC" denotes the "HalfCheetah", "Ho" denotes the "Hopper", "W2d" denotes the "Walker2d".

| Method | Random | | | Full Replay | | |
|---|---|---|---|---|---|---|
| | HC | Ho | W2d | HC | Ho | W2d |
| Normalized Score | 31.2±0.5 | 31.4±1.1 | 19.8±2.1 | 87.6±3.6 | 101.9±3.6 | 90.6±7.4 |

*Table 13.* Ablation study with initial pool size 20. "HC" denotes the "HalfCheetah", "Ho" denotes the "Hopper", "W2d" denotes the "Walker2d". Results show percentage degradation compared to the standard pool size of 100.

| Dataset | Random | | | Medium | | | Expert | | |
|---|---|---|---|---|---|---|---|---|---|
| | HC | Ho | W2d | HC | Ho | W2d | HC | Ho | W2d |
| Performance variation | ↓2.04% | ↓0.58% | ↓1.34% | ↓6.12% | ↓4.27% | ↓22.6% | ↓8.68% | ↓2.65% | ↓11.8% |

*Table 14.* Ablation study with initial pool size 20. "HC" denotes the "HalfCheetah", "Ho" denotes the "Hopper", "W2d" denotes the "Walker2d". Results show percentage degradation compared to the standard pool size of 100.

| Dataset | Medium-Expert | | | Medium-Replay | | | Full-Replay | | |
|---|---|---|---|---|---|---|---|---|---|
| | HC | Ho | W2d | HC | Ho | W2d | HC | Ho | W2d |
| Performance variation | ↓17.3% | ↓2.48% | ↓9.88% | ↓25.9% | ↓1.31% | ↓67.4% | ↓20.1% | ↓1.04% | ↓5.48% |

those in Table 11, we observe that policy performance is monotonically non-decreasing as $k$ increases, even when neural networks are used as function approximators.

### D.8. Ablation Study (RQ3)

**Effect of Convex Combination.**    The construction of the posterior belief hinges on the coefficient $\alpha$, which is typically determined by solving Eq.(5). We ablate the proposed weight averaging by comparing it to a variant that selects a single model from the pessimistic subset $\widetilde{\mathcal{T}}$. As Table 12 shows, this variant suffers significant performance degradation, confirming the effectiveness of both weight averaging and PhyB.

**Performance decline with limited initial pool.**    In the standard setting, the initial pool size is set to 100 (see D.3). We present the performance degradation observed when reducing the pool size to 20, while keeping all other hyperparameters fixed. As detailed in Table 13 and Table 14, a consistent performance drop is evident across all datasets with this reduced pool size. Notably, the degradation is more severe in complex control tasks, specifically HalfCheetah and Walker2d.

