# OpenReview forum: "Regularized Offline Policy Optimization with Posterior Hybrid Bayesian Belief"
_ICML.cc/2026/Conference — ICML 2026 regular_

### Official Review · Reviewer_bXBN · 2026-03-11

**Soundness:** 3
**Presentation:** 2
**Significance:** 3
**Originality:** 3
**Overall Recommendation:** 3
**Confidence:** 4

**Summary:**

Author proposed a new formulation Posterior Hybrid Bayesian belief (PhyB) and a corresponding update scheme (Bregman-Regularized Policy Iteration) for offline RL. Instead of using the bottom $k$-th model as pessimistic objective, PhyB creates a hybrid multimodal posterior in order to generalize pessimism level to continuous space. To further optimize policy over multimodal posterior, Bregman-Regularized Policy Iteration is proposed. Author also runs experiments on multiple benchmarks and achieves comparable results to other SOTA algorithms.

**Compliance With Llm Reviewing Policy:**

Affirmed.

**Key Questions For Authors:**

Q1.My main concern is the narrative around the proposed “posterior.” If removing the likelihood-ratio construction does not change the practical algorithm, then it is unclear to me whether Proposition 1 is a real algorithmic contribution or mainly a post hoc interpretation. If the posterior is not actually used in the algorithm, then I do not understand why we need to construct it.

Q2. I cannot find the comparison of computation cost in the paper. Can you report the cpu/gpu time of PhyB? Also, how well does the algorithm scale to larger $N$ and $k$?

**Limitations:**

yes

**Strengths And Weaknesses:**

S1. This paper provides a new way to aggregate bottom $k$ models to the objective, which also allows continuous pessimistic level.

S2. The proposed algorithm achieves impressive empirical results comparing to existing SOTA methods.

W1. Some notation in unclear:
(1) In definition 2, the pessimistic subset should depend on given (s, a) pair. Shouldn't the notation be $\tilde{\mathcal{T}}(s,a)$?
(2) In equation 5, $\sum_{i=0}^{k-1} \alpha_i = 1$?

W2. The derivation and interpretation of the proposed “posterior” are unclear.
The pessimistic subset is defined through bottom-$k$ selection based on $q_{\tau_i}(s,a)$, which is local to each \((s,a)\) pair. The paper then defines a posterior-like belief $\tilde{\mathbb P}(\tau)$ through the expectation identity in line 154, and finally constructs a likelihood ratio so that this identity holds. This makes the derivation somewhat different from my understanding in Bayesian: instead of starting from a probabilistic model and deriving the posterior, the paper seems to first specify the desired pessimistic expectation and then construct a likelihood to match it.

As a result, it is unclear what role the actual dataset plays in the posterior update, beyond its indirect influence through the learned ensemble and the resulting $q_{\tau}(s,a)$ values. Consequently, $\tilde{\mathbb P}(\tau)$ seems closer to a local reweighting rule than to a genuine posterior over dynamics.


W3. Computation cost of PhyB + Bregman-Regularized Policy Iteration is not reported in the paper.

---

> ### Author Rebuttal · Authors · 2026-03-28
>
> We sincerely appreciate the reviewer's constructive feedback and kind recognition of our paper's quality and performance. We are grateful for the opportunity to address the following questions.
>
> # Response to W1: Unclear notations.
>
> We thank the reviewer for their meticulous reading and for pointing out these notation refinements.
>
> (1) You are entirely correct. The pessimistic subset $\widetilde{\mathcal{T}}$ is indeed dynamic and depends on the specific state-action pair $(s, a)$. We agree that the notation $\widetilde{\mathcal{T}}(s, a)$ is more precise to reflect this dependency.
>
> (2) Your interpretation is correct. In Equation 5, the coefficients $\alpha_i$ are indeed constrained such that $\sum_{i=0}^{k-1} \alpha_i = 1$.
>
>
> # Response to W2: Posterior and the role of dataset.
>
> We appreciate this insightful observation. It is true that our derivation starts from a desired pessimistic expectation to construct the likelihood ratio, which may initially appear as a local reweighting rule. However, we clarify that this approach is formally grounded in **Generalized Bayesian Inference** (e.g., Gibbs posteriors).
>
> The actual dataset $\mathcal{D}$ characterizes epistemic uncertainty through the learned ensemble, where increased variance directly signals regions with limited data support. To address these regions in an offline setting, PhyB constructs a likelihood function that reweights the prior, exponentially shifting probability mass toward pessimistic models to provide robust regularization.
>
> As we highlight in the Remark (Asymptotic Behavior), our formulation is theoretically grounded. As $N \to \infty$, our likelihood ratio converges to an exponential reweighting $\xi_\infty(q_\tau) \propto \exp(-q_\tau/\lambda)$. This explicitly recovers a Gibbs posterior, where probability mass is exponentially shifted toward models with lower $Q$-values (pessimistic models).
>
> # Response to W2: The implementation of the posterior.
> The reviewer correctly identifies that a Bayesian approach requires a formal principle to derive the posterior from a probabilistic model. In PhyB, the core principle is defined by the expectation identity in Line 154:
> $$\mathbb{E}\_{\widetilde{\mathbb{P}}(\tau)}\left[q_\tau(s,a)\right]=\sum_{i=0}^{k-1}\mathbb{E}\_{\mathbb{P}(\tau)}\left[{\alpha_i}q_{\tau_i}(s,a)\right].$$
>
> This theoretical derivation directly dictates our code implementation: we sample an ensemble of $N$ models $\{\tau_0, \dots, \tau_{N-1}\}$, identify the bottom-$k$ values $\{q_{\tau_i}(s,a)\}$, and apply the weights $\alpha_i$. Mathematically, this weighted combination of the ensemble is equivalent in expectation to drawing models directly from the continuous posterior $\widetilde{\mathbb{P}}(\tau)$ and computing values ($q_{\tau}(s,a)$).
>
> In short, directly sampling models from the posterior $\widetilde{\mathbb{P}}(\tau)$ to compute the expectation is intractable. Instead, we sample from the prior $\mathbb P(\tau)$ and apply weights, thereby obtaining results equivalent to sampling from the posterior.
>
> # Response to W3: Computation cost.
>
> Like other model-based methods, PhyB's training time scales linearly with ensemble size $N$, while inference (policy sampling) is computationally inexpensive. To ensure a fair comparison, we benchmark PhyB against **model-based (PMDB, ADM)** and **model-free (DMG, TD3+BC)** baselines on the same GPU with the same batch size, while maintaining consistent dynamics model parameters for the model-based methods. We report the GPU runtime for each method, as summarized in the following.
>
> | Method | Train (ms/iteration) | Evaluation (ms/iteration) |
> | :--- | :--- | :--- |
> | PhyB ($N$=20) | 198.8428$\pm$0.9615 | 3.0978$\pm$0.0126 |
> | PhyB ($N$=10) | 106.4067$\pm$0.3879 | 3.0744$\pm$0.0139 |
> | PhyB ($N$=5) | 59.0712$\pm$0.3292 | 3.0889$\pm$0.0142 |
> | PMDB | 82.3452$\pm$0.6088 | 2.9128$\pm$0.0886 |
> | ADM | 29.7432$\pm$0.3419 | 9.0266$\pm$0.4126 |
> | DMG | 11.3725$\pm$0.1962 | 1.6785$\pm$0.0051 |
> | TD3+BC | 7.2683$\pm$0.1789 | 1.1516$\pm$0.0017 |
>
> Regarding the impact of the hyperparameter $k$ on training efficiency: the additional computational overhead is negligible. The training process involves a single forward pass for all $N$ models in the ensemble to compute the predicted values $\\{q_{\tau_i}(s,a)\\}_{i=0}^{N-1}$, followed by a standard sorting operation to select the $k$ lowest values for the convex combination. Since the forward pass (which scales with $N$) is the primary bottleneck and the subsequent sorting/selection of $k$ indices occurs in $O(N \log N)$ or $O(N)$ time on the GPU, the specific choice of $k$ has virtually no impact on the overall training wall-clock time.
>
> ---
> Should any concerns remain, we are fully committed to addressing them to the best of our ability. We sincerely appreciate your time and consideration.

---

> > ### Author Rebuttal · Reviewer_bXBN · 2026-04-05
> >
> > Thank you for your response. However, the paper’s use of the term “Bayesian” deviates from my understanding of Bayesian inference. Therefore, I will keep my score unchanged.

---

> > > ### Author Response · Authors · 2026-04-05
> > >
> > > Thank you for your hard work and time. Below is my response, and I hope it addresses your concerns:
> > >
> > > ## Bayesian RL.
> > > In modern Bayesian reinforcement learning, the model is typically treated as a random variable. Before observing the dataset, the distribution that the model follows is regarded as the prior distribution. After observing the dataset, a posterior distribution is derived based on a certain likelihood function [1].
> > >
> > > ## Reviewer’s Concern:
> > >
> > > > This makes the derivation somewhat different from my understanding in Bayesian: instead of starting from a probabilistic model and deriving the posterior, the paper seems to first specify the desired pessimistic expectation and then construct a likelihood to match it.
> > >
> > > In our method, the expectation identity in Line 154 of the paper serves as the design principle for the likelihood function:
> > > $$\mathbb{E}\_{\widetilde{\mathbb P}(\tau)} [ q\_{\tau}(s, a) ] = \sum_{i=0}^{k-1} \mathbb{E}\_{\mathbb P(\tau)} [ \alpha\_i q\_{\tau\_i}(s, a) ].$$
> > >
> > > This principle guarantees that **sampling models from the posterior to compute the expectation is equivalent to sampling from the prior and applying weights**.  We adopt this principle because direct sampling from the posterior is computationally intractable, typically requiring expensive MCMC methods, whereas the prior (e.g., a uniform distribution) is readily accessible. This principle then leads to the specific form of the likelihood function (i.e., the content of Proposition 1).
> > >
> > > Therefore, **we start with a prior distribution and derive the posterior distribution based on the likelihood function defined by the expectation identity in Line 154**. Hence, our method also follows the standard Bayesian inference procedure.
> > >
> > > ## Reviewer’s Concern:
> > > > As a result, it is unclear what role the actual dataset plays in the posterior update.
> > >
> > > The dataset $\mathcal{D}$ plays a foundational role: it is first used to train the model ensemble, which captures the epistemic uncertainty of the environment. Subsequently, the likelihood function is explicitly constructed based on the values $\\{q_{\tau}(s, a)\\}_{i=0}^{k-1}$ produced by the Pessimistic Subset (Definition 2).
> > >
> > > As established by the Proposition 1, **the likelihood depends directly on the $q_{\tau}(s,a)$. This ensures that the posterior update is intrinsically tied to the dataset, aligning with standard Bayesian principles where the data informs the shift from prior to posterior.**
> > >
> > > ---
> > > We will revise the manuscript to make this process clearer.
> > >
> > > ## Reference
> > >
> > > [1] Ni, Tianwei, Esther Derman, Vineet Jain, Vincent Taboga, Siamak Ravanbakhsh, and Pierre-Luc Bacon. "Long-Horizon Model-Based Offline Reinforcement Learning Without Conservatism." arXiv preprint arXiv:2512.04341 (2025).

---

### Official Review · Reviewer_DjyA · 2026-03-12

**Soundness:** 3
**Presentation:** 2
**Significance:** 2
**Originality:** 2
**Overall Recommendation:** 4
**Confidence:** 4

**Summary:**

This paper studies offline reinforcement learning under epistemic uncertainty and proposes a framework called Posterior Hybrid Bayesian Belief (PhyB). The key idea is to construct a pessimistic posterior distribution over an ensemble of learned dynamics models. Instead of using either a simple ensemble average or a worst-case model, the method ranks models using a value-based statistic and selects the bottom-k pessimistic models. A soft posterior is then defined over this subset using exponential weighting. The resulting belief induces a Bellman operator for value evaluation, and policies are optimized using a Bregman-regularized policy iteration procedure. The paper provides theoretical analysis including contraction properties and pessimism guarantees, and evaluates the method on D4RL benchmarks and a stochastic liquidation task. The proposed method achieves competitive or strong performance on several tasks.

**Compliance With Llm Reviewing Policy:**

Affirmed.

**Key Questions For Authors:**

(1) The hybrid posterior is defined using value-based scores derived from the critic. Is this posterior intended to represent a global belief over models, or is it a state-action dependent reweighting that changes during training?

(2) The pessimism guarantee relies on the assumption that the true dynamics model lies within the pessimistic subset with high probability. How realistic is this assumption in practical deep RL settings with learned neural models?

(3) Since the ranking of models depends on critic estimates, how sensitive is the method to inaccurate value functions early in training? Have the authors observed instability in the subset selection during learning?

(4) The theory suggests that increasing ensemble size increases pessimism. In practice, however, larger ensembles are often used to improve uncertainty estimation. How should practitioners choose the ensemble size $N$ and pessimistic subset size $k$?

**Limitations:**

The proposed approach still relies on a finite ensemble of learned models to approximate epistemic uncertainty and does not perform exact Bayesian inference. The method also depends on critic estimates to rank models, which may introduce bias if value estimates are inaccurate. Finally, the empirical evaluation is limited to a relatively small set of benchmarks, and further experiments would help clarify the robustness and generality of the approach.

**Strengths And Weaknesses:**

Strengths:

(1) The paper addresses an important problem in offline RL: how to account for epistemic uncertainty in model-based policy optimization. The proposed hybrid posterior over ensemble models is an interesting idea that provides a flexible way to interpolate between expectation-based and worst-case pessimistic methods. The formulation is conceptually appealing and attempts to connect Bayesian reasoning with practical ensemble-based approaches.

(2) Another strength is that the paper provides a unified framework combining the posterior construction, Bellman evaluation operator, and Bregman-regularized policy optimization. Theoretical results such as contraction properties and pessimistic value bounds provide some justification for the approach. Empirically, the method demonstrates promising results on standard offline RL benchmarks and a stochastic control task.

Weaknesses:

(1) The main weakness is that the Bayesian interpretation of the method is somewhat unclear. The hybrid posterior is defined using value estimates derived from the current critic and policy, meaning the induced belief depends on the learning dynamics rather than purely on data likelihood and priors. As a result, the method appears closer to a risk-sensitive reweighting of ensemble models than to true Bayesian posterior inference.

(2) Some of the theoretical guarantees also rely on assumptions that are difficult to verify in practical deep RL settings. In particular, the pessimism guarantees assume that the true model lies within the pessimistic subset with high probability. It is not clear how realistic this assumption is when models are learned from limited offline data.

(3) Another concern is that the ranking of ensemble models depends on critic estimates, which may be inaccurate early in training. This could potentially introduce instability or bias in the posterior weights. Finally, while the empirical results are promising, the evaluation could be strengthened with clearer experimental protocols and additional analysis of uncertainty calibration.

---

> ### Author Rebuttal · Authors · 2026-03-29
>
> We appreciate the reviewer's insightful comments and positive evaluation. Your support motivates us. We provide our responses below.
>
> # Response to Q1: Hybrid Posterior
> The posterior represents a global belief over the model space.
>
> Theoretically, as shown in our Remark (Asymptotic Behavior), as the ensemble size $N \to \infty$, the likelihood ratio converges to $\xi_\infty(q_\tau) \propto \exp(-q_\tau/\lambda)$. This explicitly recovers the functional form of a Gibbs posterior.
>
> In practice, since maintaining a continuous posterior is intractable, we employ a discrete ensemble to approximate the continuous distribution, while using a convex combination to estimate the expectation.
>
> # Response to Q2: On the Rationality of Assumptions in Theorem 2.
> We thank the reviewer for this insightful question. In practice, this assumption is generally satisfied for two primary reasons:
>
> First, with sufficient data coverage, it is standard to train a dynamics model with high predictive accuracy.
>
> Second, consistent with prior literature [1, 2], employing a large initial model pool is critical for performance as we construct the ensemble $\mathcal{T}$ by sampling $N$ times from this pool (See Appendix D.3). Our experimental results in the table below confirm this necessity, where reducing the initial pool size from 100 to 20 significantly degrades performance, validating the importance of maintaining a large initial model pool.
>
> | Dataset | Random (HC) | Random (Ho) | Random (W2d) | Medium (HC) | Medium (Ho) | Medium (W2d) | Expert (HC) | Expert (Ho) | Expert (W2d) |
> | :--- | :--- | :--- | :--- | :--- | :--- | :--- | :--- | :--- | :--- |
> | Performance | $\downarrow$ 2.04% | $\downarrow$ 0.58% | $\downarrow$ 1.34% | $\downarrow$ 6.12% | $\downarrow$ 4.27% | $\downarrow$ 22.6% | $\downarrow$ 8.68% | $\downarrow$ 2.65% | $\downarrow$ 11.8% |
>
> | Dataset | Medium-Expert (HC) | Medium-Expert (Ho) | Medium-Expert (W2d) | Medium-Replay (HC) | Medium-Replay (Ho) | Medium-Replay (W2d) | Full-Replay (HC) | Full-Replay (Ho) | Full-Replay (W2d) |
> | :--- | :--- | :--- | :--- | :--- | :--- | :--- | :--- | :--- | :--- |
> | Performance  | $\downarrow$ 17.3% | $\downarrow$ 2.48% | $\downarrow$ 9.88% | $\downarrow$ 25.9% | $\downarrow$ 1.31% | $\downarrow$ 67.4% | $\downarrow$ 20.1% | $\downarrow$ 1.04% | $\downarrow$ 5.48% |
>
> "HC" denotes "HalfCheetah", "Ho" denotes "Hopper", "W2d" denotes "Walker2d".
>
> # Response to Q3:  How sensitive is the method to inaccurate value functions early in training?
>
> While the subset selection depends on critic estimates, empirical results in Figure 2 and Appendix Figure 3 demonstrate that PhyB remains remarkably robust to initial value inaccuracies, with evaluation performance showing a steady, monotonic increase toward convergence.
>
> To mitigate potential instability or overestimation, we augment the training of the value function by blending real transitions from the dataset with rollouts from the learned dynamics models.
>
> Furthermore, our approach demonstrates inherent stability by using a fixed set of hyperparameters across all D4RL tasks (Appendix D.2) without per-task tuning. This confirms the reliability of our ranking mechanism, even as the critic evolves.
>
> # Response to Q4:  Ensemble Size $N$ and Pessimistic Subset Size $k$.
>
> To ensure a fair comparison, we maintain a fixed ensemble size $N=10$ and $k=5$ across all primary experiments. A moderate $N$ is vital, as the ensemble $\mathcal T$ captures our prior knowledge of the environment. Technically, for a fixed computational budget (determined by $N$), the hyperparameter $k$ directly controls the level of conservatism: a smaller $k$ (e.g., $k < N/2$) enforces stricter pessimism, which is typically beneficial for low-quality datasets (e.g., random), whereas a larger $k$ (e.g., $k \ge N/2$) allows for more optimism in higher-quality datasets.
>
> # Response to Limitation：More Empirical Evaluations.
>
> We report that PhyB achieves performance on the AntMaze-Umaze task comparable to model-free SOTA methods like DMG. This is a significant result given that AntMaze is a sparse-reward environment where model-based offline RL (e.g., MOBILE) typically struggles and remains inferior to model-free methods.
>
> | Task | AntMaze-Umaze| AntMaze-Umaze-Diverse  |
> | :--- | :--- | :---  |
> | PhyB  | 69.9$\pm$2.0 | 91.8$\pm$2.5 |
> | DMG  | 75.4$\pm$8.1 | 92.4$\pm$1.8 |
> | MOBILE | 40.3$\pm$20.4 | 84.3$\pm$3.5|
>
> ---
>
> We welcome any further questions and stand ready to offer detailed explanations whenever needed.
>
> ## References
>
> [1] Guo, K., Yunfeng, S., and Geng, Y. Model-based offline reinforcement learning with pessimism-modulated dynamics
> belief. In Advances in Neural Information Processing
> Systems, 2022.
>
> [2] Ni, Tianwei, Esther Derman, Vineet Jain, Vincent Taboga, Siamak Ravanbakhsh, and Pierre-Luc Bacon. "Long-Horizon Model-Based Offline Reinforcement Learning Without Conservatism." arXiv preprint arXiv:2512.04341 (2025).

---

> > ### Author Rebuttal · Reviewer_DjyA · 2026-04-04
> >
> > Thank you for your detailed response. I don't have further questions.

---

> > > ### Author Response · Authors · 2026-04-04
> > >
> > > Thank you for your efforts and support in the review process.

---

### Official Review · Reviewer_Ni4r · 2026-03-12

**Soundness:** 3
**Presentation:** 3
**Significance:** 3
**Originality:** 3
**Overall Recommendation:** 4
**Confidence:** 2

**Summary:**

This paper introduces Posterior Hybrid Bayesian Belief (PhyB) for offline reinforcement learning. To overcome the computational intractability of posterior inference over continuous model spaces, it reformulates the expectation as a convex combination over a pessimistic subset of candidate dynamics models. The authors propose a Bregman-regularized policy optimization algorithm to handle the multimodal posterior geometry, supported by theoretical guarantees and strong empirical results on standard benchmarks.

**Compliance With Llm Reviewing Policy:**

Affirmed.

**Final Justification:**

Weak Accept

**Key Questions For Authors:**

Q1: How is this inversion practically approximated in your code (e.g., K-FAC, conjugate gradients)? What is the wall-clock time per update compared to standard SAC/TD3?

Q2: How much does performance degrade if the initial pool of models is restricted to 15 or 20 due to compute constraints?

Q3: Since $\lambda$ directly controls the weight distribution of the pessimistic subset, how sensitive is the final normalized score to variations in $\lambda$?

**Strengths And Weaknesses:**

#### **Strengths**
* The paper provides solid proofs establishing that the Hybrid Belief Bellman Evaluation Operator is a contraction and that iteratively solving the Bregman-regularized subproblems guarantees monotonic policy improvement.
* By maintaining a multimodal posterior belief, PhyB avoids the over-conservatism of standard robust MDPs and allows for continuous, data-driven modulation of pessimism.
* The method demonstrates superior or competitive performance on D4RL and shows exceptional robustness in the highly stochastic optimal liquidation task where strong baselines fail.

#### **Weaknesses**
* The policy update step relies on a natural gradient-style update (Eq. 11), requiring the computation and inversion of the Hessian $[\nabla^2 \psi(\phi_t)]^{-1}$. For deep neural networks, exact Hessian inversion is computationally intractable.

* The method requires independently pre-training 100 dynamics models via maximum likelihood estimation to sample an ensemble of $N=10$. This is massively more expensive than standard model-based offline RL.


* While hyperparameters $\omega$ and $(N, k)$ are thoroughly ablated, there is no sensitivity analysis for the entropy coefficient $\lambda$ (set to 0.33).

---

> ### Author Rebuttal · Authors · 2026-03-28
>
> We truly appreciate the time and effort you have dedicated to providing such insightful feedback.
>
> # Response to Q1: Implementation of the inversion.
> You are correct that directly inverting the weight matrix in Eq. (11) is computationally prohibitive in deep RL. In practice, we solve this by implementing the policy update via mirror descent:$$\tilde{\theta}\_{k+1} = \arg \max\_{\theta \in \Theta} \{ \langle g_k, \theta \rangle - \beta D\_{\psi}(\theta, \theta_k) \},$$
> $$\theta\_{k+1} = \theta\_k + \eta\_k(\tilde{\theta}\_{k+1} - \theta\_k).$$
> where $g_k$ is the gradient, $\eta\_k$ is the learning rate and $D\_{\psi}$ is the Bregman divergence.
>
> This iterative process is a theoretically sound approximation of the closed-form solution in Eq. (11). We will clarify this implementation detail and its theoretical equivalence in the Appendix C.
>
> # Response to Q1: Computational overhead
> Like other model-based methods, PhyB's training time scales linearly with ensemble size $N$, while inference (policy sampling) is computationally inexpensive. We benchmark PhyB against **model-based (PMDB, ADM) and model-free (DMG, TD3+BC,SAC)** baselines on the same GPU with the same batch size, while maintaining consistent dynamics model parameters for the model-based methods. We report the GPU runtime for each method, as summarized in the following.
>
> | Method | Train (ms/iteration) | Evaluation (ms/iteration) |
> | :--- | :--- | :--- |
> | PhyB ($N$=20) | 198.8428$\pm$0.9615 | 3.0978$\pm$0.0126 |
> | PhyB ($N$=10) | 106.4067$\pm$0.3879 | 3.0744$\pm$0.0139 |
> | PhyB ($N$=5) | 59.0712$\pm$0.3292 | 3.0889$\pm$0.0142 |
> | PMDB | 82.3452$\pm$0.6088 | 2.9128$\pm$0.0886 |
> | ADM | 29.7432$\pm$0.3419 | 9.0266$\pm$0.4126 |
> | DMG | 11.3725$\pm$0.1962 | 1.6785$\pm$0.0051 |
> | TD3+BC | 7.2683$\pm$0.1789 | 1.1516$\pm$0.0017 |
> | SAC | 11.8052$\pm$0.4927 | 1.9827$\pm$0.0919 |
>
> TD3+BC incorporates a behavior cloning regularizer into the TD3 objective, so its training overhead should be comparable to that of TD3.
>
> # Response to Q2: Performance degradation if the initial pool of models is 20.
> We conduct an **ablation study by reducing the pool size from our default of 100 to 20**, keeping all other hyperparameters fixed. We report the performance degradation relative to the PhyB across various datasets.
>
> As shown in the table following, a consistent performance drop is evident across all datasets with this reduced pool size. Notably, the degradation is more severe in complex control tasks, specifically HalfCheetah and Walker2d.
>
> | Dataset | Random (HC) | Random (Ho) | Random (W2d) | Medium (HC) | Medium (Ho) | Medium (W2d) | Expert (HC) | Expert (Ho) | Expert (W2d) |
> | :--- | :--- | :--- | :--- | :--- | :--- | :--- | :--- | :--- | :--- |
> | Performance | $\downarrow$ 2.04% | $\downarrow$ 0.58% | $\downarrow$ 1.34% | $\downarrow$ 6.12% | $\downarrow$ 4.27% | $\downarrow$ 22.6% | $\downarrow$ 8.68% | $\downarrow$ 2.65% | $\downarrow$ 11.8% |
>
> | Dataset | Medium-Expert (HC) | Medium-Expert (Ho) | Medium-Expert (W2d) | Medium-Replay (HC) | Medium-Replay (Ho) | Medium-Replay (W2d) | Full-Replay (HC) | Full-Replay (Ho) | Full-Replay (W2d) |
> | :--- | :--- | :--- | :--- | :--- | :--- | :--- | :--- | :--- | :--- |
> | Performance  | $\downarrow$ 17.3% | $\downarrow$ 2.48% | $\downarrow$ 9.88% | $\downarrow$ 25.9% | $\downarrow$ 1.31% | $\downarrow$ 67.4% | $\downarrow$ 20.1% | $\downarrow$ 1.04% | $\downarrow$ 5.48% |
>
> # Response to Q3: Sensitivity analysis of the $\lambda$.
> The hyperparameter $\lambda$ governs the concentration of the posterior belief: a smaller $\lambda$ shifts more weight toward pessimistic models, while a larger $\lambda$ leads to a more uniform distribution. Sensitivity analysis shows: (1). Performance remains robust across a reasonable range of $\lambda$. (2). Lower-quality datasets benefit from smaller $\lambda$ to mitigate OOD risks. Conversely, in high-quality datasets, an overly small $\lambda$ can be excessively conservative, leading to suboptimal performance.
>
> | Dataset | Random (HC) | Random (Ho) | Random (W2d) | Medium (HC) | Medium (Ho) | Medium (W2d) |
> | :--- | :--- | :--- | :--- | :--- | :--- | :--- |
> | $\lambda=0.2$ | $\uparrow$ 2.95% | $\downarrow$ 0.08% | $\downarrow$ 4.81% | $\uparrow$ 3.34% | $\downarrow$ 0.46% | $\uparrow$ 0.22% |
> | $\lambda=1$ | $\uparrow$ 0.20% | $\uparrow$ 0.06% | $\downarrow$ 5.42% | $\uparrow$ 0.93% | $\uparrow$ 0.21% | $\uparrow$ 2.93% |
>
> | Dataset | Expert (HC) | Expert (Ho) | Expert (W2d) | Medium Expert (HC) | Medium Expert (Ho) | Medium Expert (W2d) |
> | :--- | :--- | :--- | :--- | :--- | :--- | :--- |
> | $\lambda=0.2$ | $\downarrow$ 0.56% | $\downarrow$ 1.45% | $\downarrow$ 0.44% | $\downarrow$ 1.22% | $\uparrow$ 1.43% | $\downarrow$ 0.82% |
> | $\lambda=1$ | $\uparrow$ 2.73% | $\uparrow$ 1.13% | $\uparrow$ 4.79% | $\uparrow$ 2.18% | $\downarrow$ 0.49% | $\downarrow$ 0.17% |
>
> "HC" denotes "HalfCheetah", "Ho" denotes "Hopper", "W2d" denotes "Walker2d".
>
> Please let us know if you have any further questions.

---

> > ### Author Rebuttal · Reviewer_Ni4r · 2026-04-01
> >
> > Thank you for your response. I will work on improving my score accordingly.

---

> > > ### Author Response · Authors · 2026-04-02
> > >
> > > We would like to express our sincere gratitude for your hard work and dedication during the review process.

---

### Official Review · Reviewer_gdvc · 2026-03-16

**Soundness:** 2
**Presentation:** 2
**Significance:** 3
**Originality:** 3
**Overall Recommendation:** 4
**Confidence:** 3

**Summary:**

This paper focuses on the offline RL problem, combining conservatism with a Bayesian approach. It aims to overcome structural assumptions on the prior belief while maintaining tracktability. The proposed method, PhyB, is to take a finite subset of models playing the role of quantiles, each corresponding to a different performance level. Then, a convex combination between these models is optimized. A performance bound is established between the resulting value and the theoretical one. Experiments on D4RL compare PhyB with offline RL baselines.

**Compliance With Llm Reviewing Policy:**

Affirmed.

**Final Justification:**

The authors clarified my misunderstandings and answered my questions.

**Key Questions For Authors:**

My questions appear in the previous comments.

**Limitations:**

The authors do not discuss the limitations of this work. For example, what is the computational overhead of constructing $\tilde{\mathcal{T}}$ at each iteration?

**Strengths And Weaknesses:**

**Strengths**

- The paper is well-written and structured.
- The performance resulting from PhyB seems promising.

**Weaknesses**
I am a bit skeptic about the theory supporting PhyB. The flow is unclear, assumptions or structural methodology are not explicitly stated, which confuses me about the contributions of this work. I detail more below.

- Notations are inconsistent: $\tau$ for transitions becomes $\tau^*$, $\mathcal{T}$ becomes $T$, which renders the reader's understanding difficult.
- In Sec. 2.2, the authors seem to assume that the number of models is finite. I guess it is not the case. From my understanding, they are taking a model ensemble approach, each ensemble corresponding to a quantile level on the (possibly continuous) set of models. The text is very ambiguous in that respect, and I may be misunderstanding.
- Def. 1 is unclear: Why is there an $s,a$ subscript in $\mathcal{R}_{s,a}$? Why is the return nested with respect to transition models? In finite horizon BAMDPs, I would understand that a new model is sampled at the beginning of each episode, but here, it seems to be sampled at each step of an infinite-horizon process.
- In Eq. (4) I have the same confusion: is the expectation taken with respect to $\mathbb{P}(\tau)$, the **joint** distribution of stepwise transitions $\tau_1, \tau_2,\cdots$?
- Capital $T$-s in Theorem 2 and the following remark should be $\tau$-s, I guess.
- l. 69: "Iterative Regularized"
- Related work [1] takes a Bayesian approach too, similarly as PhyB.

[1] Ni, Tianwei, Esther Derman, Vineet Jain, Vincent Taboga, Siamak Ravanbakhsh, and Pierre-Luc Bacon. "Long-Horizon Model-Based Offline Reinforcement Learning Without Conservatism." arXiv preprint arXiv:2512.04341 (2025).

---

> ### Author Rebuttal · Authors · 2026-03-28
>
> We thank the reviewer for the constructive feedback and recognition of our paper's quality and performance. Below are our responses to your questions.
>
> # Response to Q1: Notations are inconsistent.
>
> The symbols $T$ and $\tau$ in Theorem 2 refer to the same concept, namely the dynamics model.  The symbol $\mathcal{T}$ refers to the model ensemble. We will revise this for clarity and apologize for the confusion caused. We have also corrected the typos in the manuscript.
>
> # Response to Q2: Sec 2.2.
>
> We clarify that while the true Bayesian posterior is defined over a continuous space ,the finite set $\mathcal T$ serves as a practical ensemble approximation to handle this intractability. We will revise the text to explicitly state that  $\mathcal T$  consists of $N$ samples approximating the continuous distribution.
>
> Our critique of the **Dirac likelihood** assumption applies to how current methods utilize this finite ensemble $\mathcal T$ . Most existing works select only the single $k$-th worst model, whereas PhyB leverages the entire ensemble by considering a convex combination over these plausible models.
>
> A prime example is Robust MDP, which selects the worst-case model. This is equivalent to optimizing the **$\frac{1}{N}$-quantile of the objectives $\\{\eta(\pi, \tau)\\}_{\tau \in \mathcal{T}}$**.
>
> # Response to Q3: Def.1.
>
> We apologize for the lack of clarity in Def.1. Our intention is to define the cumulative discounted return $\mathcal{R}_{s,a}$ as a mapping from the state-action space to a real-valued performance metric.
>
> Our formulation of $\eta(\pi)$ (Eq. (3)) is designed for maximum generality: by sampling a transition model $\tau_t$ at each timestep to fit the current $(s_t, a_t)$, we capture the most flexible transition uncertainty.
>
> We emphasize that if the posterior $\widetilde{\mathbb{P}}(\tau)$ is a **point mass** or if a model is **sampled only once at the beginning of each episode and held fixed** (the standard BAMDP setting), our definition naturally reduces to the exact case you described. We will clarify this mapping and the reduction to the episodic-sampling case in the revised manuscript.
>
> # Response to Q4: Eq. (4).
>
> Your interpretation is correct. $\mathbb{P}(\tau)$ represents the distribution governing the transition model $\tau$, which can be understood as the joint distribution of stepwise transitions $\tau_1, \tau_2, \dots$ as you suggested.
>
> To clarify the relationship with standard RL: if we treat the transition model $\tau$ in Eq. (4) as a fixed constant rather than a random variable, the expression naturally reduces to the standard Bellman operator:$$\hat{\mathcal{B}}^\pi Q(s,a) = r(s,a) + \gamma\mathbb{E}_{s' \sim \tau, a' \sim \pi}[Q(s',a')].$$
>
> In our Bayesian framework, Eq. (4) generalizes this by accounting for the epistemic uncertainty embedded in $\mathbb{P}(\tau)$. We will explicitly state this connection in the revised manuscript to avoid any ambiguity regarding the expectation.
>
> #  Response to Q5:  Comparison with Ni et al. 2026.
> We thank the reviewer for highlighting this recent and relevant work. While both papers adopt a Bayesian framework by treating the dynamics model as a random variable, our core motivations and mechanisms differ significantly: Ni et al. 2026 focus on mitigating compounding errors in long-horizon rollouts by using a Bayesian-derived truncation function to ensure model-consistent transitions.
>
> In contrast, PhyB leverages a finite ensemble to approximate a continuous probability distribution over models. We specifically utilize a convex combination of these models to approximate the expected return under epistemic uncertainty, providing a practical way to balance conservatism and performance.
>
> We appreciate the suggestion and will include Ni et al. 2026 in the related work section of our revised manuscript.
>
> #  Response to Q6:  Computational overhead.
> Like other model-based methods, PhyB's training time scales linearly with ensemble size $N$, while inference (policy sampling) is computationally inexpensive. To ensure a fair comparison, we benchmark PhyB against **model-based (PMDB, ADM)** and **model-free (DMG, TD3+BC)** baselines on the same GPU with the same batch size, while maintaining consistent dynamics model parameters for the model-based methods. We report the GPU runtime for each method, as summarized in the following.
>
> | Method | Train (ms/iteration) | Evaluation (ms/iteration) |
> | :--- | :--- | :--- |
> | PhyB ($N$=20) | 198.8428$\pm$0.9615 | 3.0978$\pm$0.0126 |
> | PhyB ($N$=10) | 106.4067$\pm$0.3879 | 3.0744$\pm$0.0139 |
> | PhyB ($N$=5) | 59.0712$\pm$0.3292 | 3.0889$\pm$0.0142 |
> | PMDB | 82.3452$\pm$0.6088 | 2.9128$\pm$0.0886 |
> | ADM | 29.7432$\pm$0.3419 | 9.0266$\pm$0.4126 |
> | DMG | 11.3725$\pm$0.1962 | 1.6785$\pm$0.0051 |
> | TD3+BC | 7.2683$\pm$0.1789 | 1.1516$\pm$0.0017 |
>
> ---
> If there are any remaining concerns, we are fully committed to addressing them to the best of our ability. We appreciate your consideration.

---

> > ### Author Rebuttal · Reviewer_gdvc · 2026-04-03
> >
> > The authors addressed my concerns and clarified my misunderstandings. I raise my score to 4

---

> > > ### Author Response · Authors · 2026-04-04
> > >
> > > Thank you for the time and effort you have devoted to the review process！

---

### Decision · Program_Chairs · 2026-04-30

**Decision:**

Accept (regular)

**Comment:**

The proposed method, Posterior Hybrid Bayesian Belief (PhyB), provides a unified quantification of epistemic uncertainty through a multimodal posterior belief. The paper is well-supported by solid theoretical analysis that are matched by strong empirical results across various standard benchmarks, notably demonstrating superior performance on D4RL and exceptional robustness in high-stochasticity tasks such as optimal liquidation. While the technical merits and empirical gains are clear, the conceptual framing of the contribution was found to be somewhat misleading during the review process; specifically, the authors must clarify how the Bayesian aspect of the method fundamentally differs from an alternative reweighting scheme, as this distinction has critical consequences for the novelty claims, literature positioning, and the interpretation of the "posterior" in a traditional Bayesian sense. Addressing this conceptual ambiguity and refining the related discussion in the camera-ready version will ensure the work is correctly situated within the literature and that its unique methodological contributions are fully understood.